# Mimicked synthetic ribosomal protein complex for benchmarking crosslinking mass spectrometry workflows

Manuel Matzinger [1,3 ✉], Adrian Vasiu[1,3], Mathias Madalinski[1], Fränze Müller [1], Florian Stanek[1] & Karl Mechtler [1,2 ✉]

Cross-linking mass spectrometry has matured to a frequently used tool for the investigation of protein structures as well as interactome studies up to a system-wide level. The growing community generated a broad spectrum of applications, linker types, acquisition strategies and specialized data analysis tools, which makes it challenging to decide for an appropriate analysis workflow. Here, we report a large and flexible synthetic peptide library as reliable instrument to benchmark crosslink workflows. Additionally, we provide a tool, IMP-X-FDR, that calculates the real, experimentally validated, FDR, compares results across search engine platforms and analyses crosslink properties in an automated manner. We apply the library with 6 commonly used linker reagents and analyse the data with 6 established search engines. We thereby show that the correct algorithm and search setting choice is highly important to improve identification rate and reliability. We reach identification rates of up to ~70 % of the theoretical maximum (i.e. 700 unique lysine-lysine cross-links) while maintaining a real false-discovery-rate of <3 % at cross-link level with high reproducibility, representatively showing that our test system delivers valuable and statistically solid results.

---

[1] Institute of Molecular Pathology (IMP), Vienna BioCenter (VBC), Vienna, Austria. [2] Institute of Molecular Biotechnology, Austrian Academy of Sciences, Vienna BioCenter (VBC), Vienna, Austria. [3]These authors contributed equally: Manuel Matzinger, Adrian Vasiu. ✉email: manuel.matzinger@imp.ac.at; karl.mechtler@imp.ac.at

The field of cross-linking mass spectrometry has matured and now represents a frequently used technique for the investigation of protein structures as well as to freeze (transient) protein-protein interactions and uncover whole interactomes on a system wide level. Numerous reviews already summarized successful applications but also limitations of this technique[1–4]. The growing community also participated in the generation of a wide variety of cross-linker reagents bearing chemical reactivities mainly towards lysine (e.g. via N-Hydroxysuccinimide esters[5,6]) but also towards acidic amino acids (e.g. by amide formation[7] or hydrazines[8]), cysteine (e.g. via maleimides[9,10]) or even without any specificity (e.g. via diazirine groups[11]). With a focus on proteome wide studies and in vivo cross linking, MS-cleavable linkers, like DSSO[12] or DSBU[13], are facilitating data analysis by generating characteristic doublet ions and became commonly used. Aiming to dig deeper in the inter-actome of complex samples, reagents bearing an affinity tag for selective enrichment of cross-linked peptides were further developed[14–16]. The optimization of cross-linker specific acqui-sition strategies[17] and most recently the implementation of ion-mobility[18,19] or FAIMS filtering[20] as additional separation technique further boosted the number of possible crosslink (XL) identifications.

The broad spectrum of applications, linker types and acquisi-tion strategies[4] led to the development of lots of specialized data analysis tools[21] which makes it challenging, especially for new-comers, to decide for an appropriate analysis workflow.

Therefore, a synthetic peptide library as previously published by our group[22] is a valuable tool for standardization and can be used as a basis to decide for the optimal analysis tool in dependency of the used crosslinker and acquisition strategy. The previous peptide library was based on 95 synthetic peptides of the protein Cas9.

In this study we present a significantly improved and extended peptide library that now contains a total of 141 peptides from 38 different proteins of the E. coli ribosomal complex. This enables finding inter- and intra-protein cross-links in our results. Fur-thermore, the number of theoretical correct cross-link combina-tions is increased from 426 in the previously published version to up to 1018 in this library. In conclusion a more reliable and, if supported by the data analysis tool, separate inter/intra false discovery rate (FDR) calculation can be performed. In contrast to our previously published library system of Cas9, the peptides were now combined to 3 different libraries designed to be com-patible not only with lysine but also with aspartic- and glutamic-acid reactive cross-linkers as well as for crosslinkers bearing an azide as affinity tag, respectively. With the here reported peptide library, we mimic a real protein complex and a system that is appropriate to find optimal settings for real biological samples as well as to benchmark different crosslinker types and data analysis tools. To increase the usability of that library, we addi-tionally created a tool, IMP-X-FDR, that is capable to check the target-decoy based FDR estimation given by search engines and instead outputs the real, experimentally validated, FDR. Addi-tionally, the tool can correct the number of cross-link IDs to a real FDR of 1 or 5% by applying a score-cutoff as well as to compare the results obtained from several search engines or cross-linkers in Venn Diagrams. IMP-X-FDR completes this task in an automated manner and includes an easy-to-use graphical user interface, which broadens the potential user group. IMP-X-FDR is free to use and can be downloaded from Github (https://github.com/fstanek/imp-x-fdr[23]).

## Results
A schematic workflow of this study is shown in Fig. 1, briefly: We synthesized 141 peptides based on sequences from 38 proteins of

the E. coli ribosomal complex (Supplementary Data 1). They are designed to contain exactly one crosslink-able position. Peptides are grouped to 6–10 peptides and crosslinked groupwise. After that, all groups are pooled to obtain the crosslinked library were links between peptides of different groups or to not synthesized peptides are known false positives. The main library consists of 100 peptides containing exactly one crosslink-able lysine residue. All peptides start with the sequence WGGGGR- and their N-termini are protected by an acetate group to hinder any crosslink reaction at this position. Tryptophan thereby facilitates photometric quantitation of peptides after synthesis. C-terminal lysine residues are modified to an azide (instead of an amine) to again block the crosslink reaction. During sample processing the protected N-terminal sequence part is removed by tryptic diges-tion and azide modified lysine's are reduced to amines yielding ordinary tryptic peptides with a known crosslink position for MS/MS analysis. We additionally compiled a library not containing any azide protected lysine residue but instead exclusively those 64 peptides of the main library ending with arginine. This "enrich-able library" is compatible with azide-based affinity enrichment as done with the reagent azide-tagged acid-cleavable dis-uccinimidylbissulfoxide, (DSBSO). Finally, a third library, made from 43 peptides, is designed to contain exactly one reactive aspartic-acid or glutamic-acid for use with crosslinker reagents reactive to carboxylic acids. In this "acidic library" the C-terminal peptide part is amide protected and all sequences end as GGGG after a K or R which will again release ordinary tryptic peptides after digestion.

**Benchmarking crosslink search engines with linkers targeting lysine.** To benchmark commonly used crosslink search-algorithms we applied the MS cleavable linker reagents disuccinimidyl sulfoxide (DSSO), ureido-4,4′-dibutyric acid bis(hydroxysuccinimide) ester (DSBU) and 1,1'-carbonyldiimi-dazole (CDI) to the main library (Supplementary Data 2). As representatively shown on the data generated with DSSO the benchmarked search engines all output higher experimentally validated FDRs than the estimated 1% on crosslink level (Fig. 2A). For this dataset MS Annika[24] and MaxLynx[25] perform best, both by means of correct FDR estimation as well as by means of unique ID numbers. We additionally applied post-score-cutoffs to correct the experimentally validated FDR to ≤1%. The obtained results are in line with minimal scores recommended by the software developers (i.e. scores >100 are considered as good for MeroX[26], 50 is default and 75 seems reliable from our data; 40 is default for XlinkX[27], 41 seems reliable from our data). Although using a score-cutoff is an effective strategy to correct for accep-table FDR, our data also shows, that built in (usually target-decoy based) FDR estimations are not sufficient yet. Especially when using pLink 2[28], we had the impression that (score-based) separation of correct and incorrect IDs does not work properly meaning that the majority of crosslink IDs is lost upon applying our FDR correction. Of note, pLink 2 was initially designed to work with non-MS-cleavable linker reagents and is not optimized for HCD data in combination with cleavable linkers, which might explain its weak performance in this dataset compared to all other tested algorithms.

Instead of using score cutoff values, the comparison of identified crosslinks with more than one search engine can significantly contribute to improve the confidence in results (Fig. 2C). Using our in house-developed tool IMP-X-FDR we visualized the overlap of search results obtained from MS Annika and a second search engine and calculated the FDR in an automated manner (Examples of output Figures automatically created by IMP-X-FDR are shown in Supplementary Figs. 5 and 6). The fraction of commonly

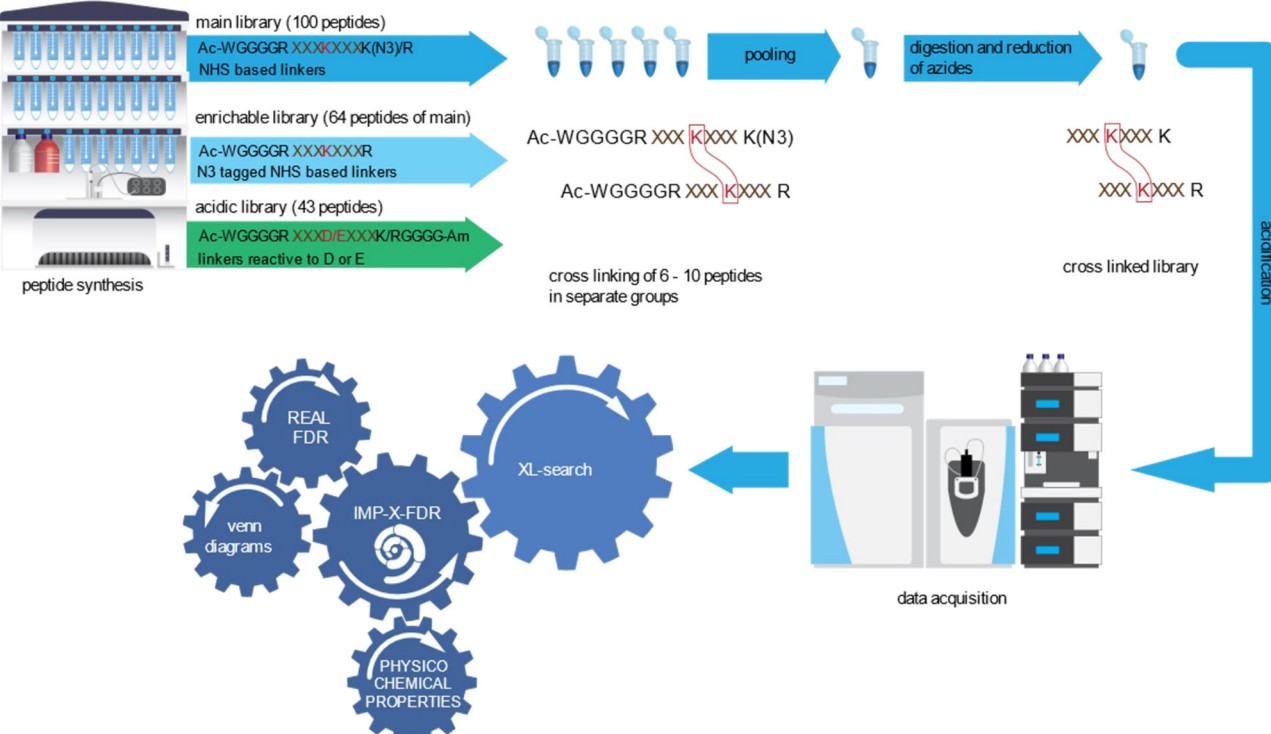

**Fig. 1 Schematic workflow of study design, from peptide synthesis.** The figure illustrates each step of the workflow, stating with synthesis of protected peptides, followed by grouping into mixes of 6-10 peptptides. Corss-linker reagents are separatley added to each group. After quenching, peptides are digested and reduced to remove all protection groups. All peptides groups are pooled into a single vial to generate the final library that is injected to LC-MS. After data analysis with a XL-search engine, post processing is performed using 10.1038/s41467-022-31701-w IMP-X-FDR.

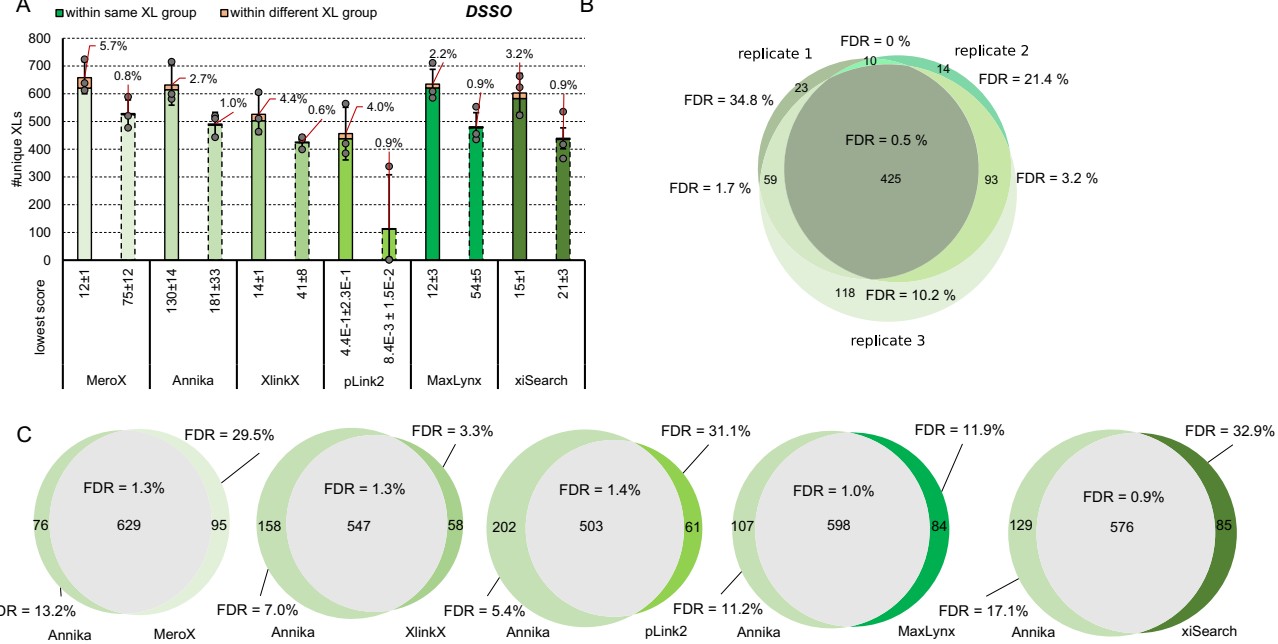

**Fig. 2 Benchmarking of data analysis tools on the example of DSSO. A** Average crosslink numbers using DSSO after acquisition using a stepped HCD MS2 method. Applied to the main library using the algorithms MeroX[26], MS Annika[24], XlinkX[27], pLink 2[28], MaxLynx[25] or xiSearch[36, 44] for analysis. All results were obtained at 1% estimated FDR (solid line bars) and corrected by applying a post-score cutoff to reach an experimentally validated FDR < =1 (dashed line bars). The experimentally validated FDR is shown as callout, error-bars indicate standard deviations of average values, $n = 3$ independent samples acquired on different days. **B** Overlap of crosslinks identified in each replicate using MS Annika. **C** Overlap of cross links identified in replicate 3 after analysis using MS Annika or an alternative algorithm as given. **B**, **C**: Experimentally validated FDRs for commonly found and exclusively found links are given. Source data are provided as a Source Data file.

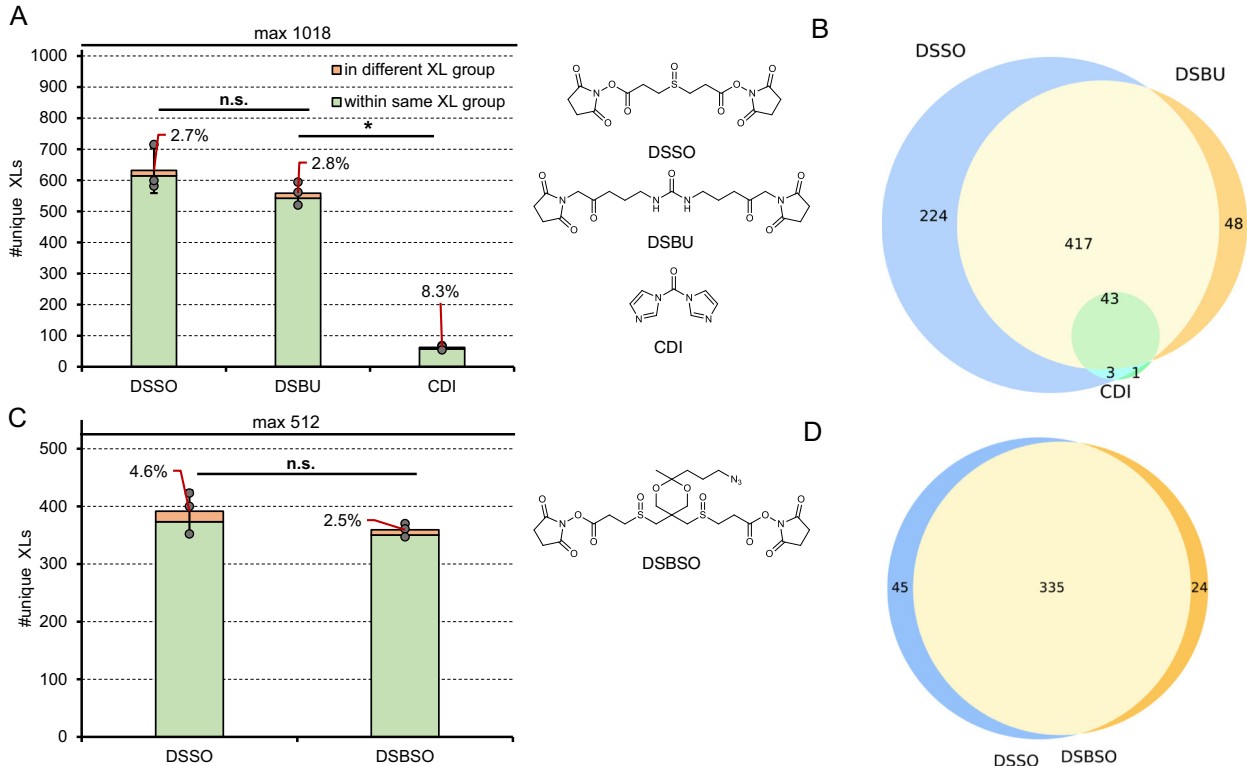

**Fig. 3 Benchmarking the linker reagents DSSO, DSBU, CDI and DSBSO.** Average number of unique crosslink IDs after acquisition using a stepped HCD MS2 strategy and maximal theoretical number of true link combinations when applying the indicated crosslinkers to the main library (**A**) or the enrichable library (**C**) and data analysis using MS Annika at 1% estimated FDR. The experimentally validated FDR is shown as callout, error-bars indicate standard deviations of average values, $n = 3$ independent samples acquired on different days, unpaired Student's t test, two tailed, $\alpha = 0.05$, * $P < 0.05$, n.s. = not significant. Obtained $P$ values: DSSO vs DSBSO: 0.1947, DSBSO vs CDI: < 0.0001, DSSO vs DSBSO: 0.2149 (**B** and **D**) Overlap of identified links from one representative replicate of (**A**) (in **B**) or (**C**) (in **D**) respectively. Source data are provided as a Source Data file.

identified unique crosslinks contains up to 629 entries (MS Annika + MeroX) and within this fraction the experimentally validated FDR is ≤1.4 % in all cases and therefore very close to the accepted 1%. On the contrary those crosslinks exclusively identified by only one search engine contain most false positives yielding to FDR rates of up to 31%. A similar effect is also observed for replicate measurements (Fig. 2B). Of 425 unique crosslinks commonly found in three replicates only 2 (0.5 %) were incorrect. While using crosslinks commonly found across replicate measurements seems to yield highly confident results, the accumulation of IDs from several replicate measurements to boost link numbers is prone to also accumulate wrong hits and should therefore be avoided. We further investigated those two crosslinks that were incorrectly assigned in all three replicates using MS Annika (Fig. 2B): The first one is a homeotypic link of the peptide MAKLTK that does not exist in the library (but in the database used to search the files). A peptide with the sequence MAKTIK of the same mass is however part of our library and was therefore very likely generating the wrongly annotated spectra. The second one connects two existing peptides (LSYDTEASIAKAK- VAVI- KAVR) that are however within different groups.

In a next step we benchmarked the reagents DSSO, DSBU and CDI on the main library (Fig. 3A, B). Expectedly, the performance of DSSO and DSBU is on a similar level, since both have comparable spacer lengths of 10.1 and 12.5 Å respectively and the same reactive site. The two linkers bear different reactive groups for MS based fragmentation which might lead to the assumption that differences in spectra quality explain the slight difference in unique link numbers. Notably this effect is software specific. MS Annika performs very well with DSSO and scores

DSSO crosslinks better than DSBU links (average score 279 for all DSSO links vs 269 for all DSBU links from our main library). In contrast MeroX performs very well with DSBU and scores those links slightly better (average score 131 for all DSSO links vs 133 for all DSBU links from our main library). In conclusion, when comparing MeroX results, DSBU (767 links on average) outperforms DSSO (658 links on average) in terms of unique crosslinks (data shown in Supplementary Data 2).

The zero length crosslinker CDI yielded in ≤80 unique crosslinks identified with all tested algorithms. This low number might be reasoned by no "real" interaction sites within the peptide library that relies on crosslink connections formed between freely moving peptides in solution. Therefore, the likelihood of two peptides being connected by a crosslinker with a very short spacer is lowered compared to those linkers with a 10–12 Å spacer. A full list of unique link numbers and experimentally validated FDRs using all tested algorithms can be found in Supplementary Data 2.

Next, we compared detectability of DSSO vs DSBSO using the enrichable peptide library (Fig. 3C, D). In this artificial system any potential steric hindrance of the azide tag of DSBSO can be neglected, hence we assume that differences in observed crosslinks are reasoned mainly by the ionizability of the resulting connected peptides. As illustrated in the Venn diagram in Fig. 3D, the overlap of identified crosslinks is indeed very high and could not be distinguished to an overlap of replicate measurements from the same linker (compare to Fig. 2B). DSSO only slightly, and non-significantly, outperforms DSBSO by means of crosslink numbers indicating a slightly increased reactivity or ionizability.

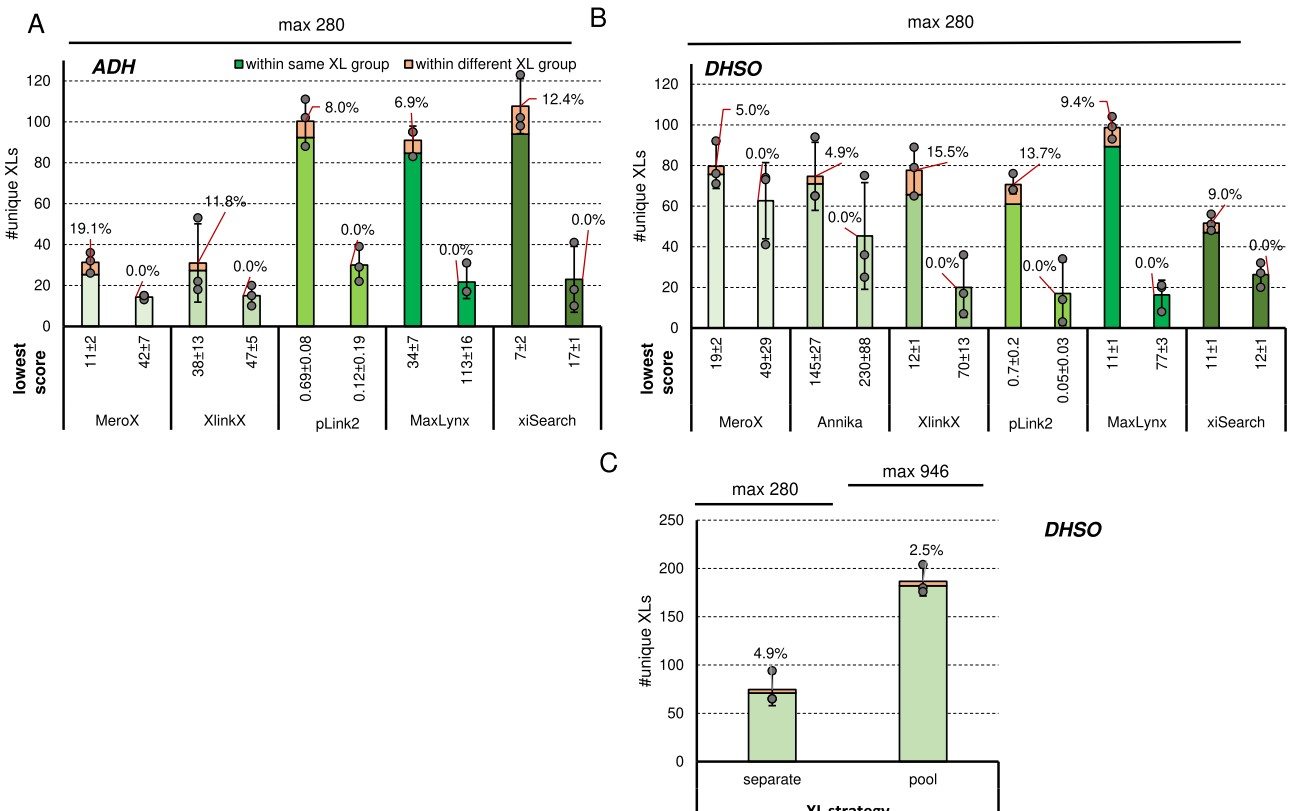

**Fig. 4 Benchmarking linker reagents reactive to acidic amino acids.** Average number of unique link IDs and maximal theoretical number of true crosslink combinations after acquisition using a stepped HCD MS2 strategy when using ADH (**A**) or DHSO (**B**) to crosslink the acidic library. Data analysis was performed using the indicated algorithm at 1% estimated FDR (solid line bars) and corrected by applying a post-score cutoff to reach an experimentally validated FDR < = 1 (dashed line bars). **C** As B but when crosslinking the library either in separate groups or adding the linker to a pool of all peptides to boost resulting ID numbers. Data analysis using MS Annika at 1% estimated FDR. **A–C** The experimentally validated FDR is shown as callout, error-bars indicate standard deviations of average values, $n = 3$ independent samples acquired on different days. Source data are provided as a Source Data file.

**Benchmarking crosslink search engines with carboxylic acid reactive linkers**. Next, we investigated two reagents targeting acidic amino acids: The non-cleavable adipic acid dihydrazide (ADH) and the cleavable dihydrazide sulfoxide (DHSO) (Fig. 4). These linkers were applied to a smaller peptide library with a reduced number of only 280 theoretically possible crosslinks formed, however, less than 40 % of this number was identified in all cases. This indicates a lowered reaction efficiency compared to the more established NHS ester-based linkers, where more than 60% of the theoretical crosslink number was reached (Figs. 1 and 2).

For the non-cleavable ADH linker, pLink 2 and MaxLynx seem to perform significantly better, both by means of reliability and ID numbers, than their competitors. However, calculated experimentally validated FDR values seem extraordinarily high for both reagents and every software tested on the acidic library. A proper FDR calculation might be hindered by the relatively small number of crosslinks available in this system.

Of note, 4-methylmorpholinium chloride (DMTMM), that was used as coupling reagent for ADH and DHSO, could form zero-length connections between amines and carboxylic acids. However, only two synthetic peptides of the acidic library contain any lysine residue except for those that are terminal after tryptic cleavage. We investigated the presence of DMTMM crosslinked (undigested) peptides and found no evidence for such a side reaction. The low number of crosslink identifications in the acidic library might be reasoned by a slow reaction kinetics and the fact that two steps (activation of carboxylic acids by DMTMM followed by nucleophilic attack of the hydrazine group) are required instead of only one as is the case for NHS based reagents. To boost the number of crosslink IDs we further tested DHSO on a pool of all 41 peptides of the acidic library. This increases the number to possible crosslink combinations from 280 to 946 and therefore close to the value of the main library. The number of identified crosslinks maintained low at ~20% of the theoretical maximum when using MS Annika (Fig. 4C).

To better understand the reaction chemistry of these hydrazine-based linkers, we analyzed the results obtained for DHSO using our in house developed tool IMP-X-FDR to investigate the distribution of amino acids in detected crosslink-sequence-matches (CSMs) (Supplementary Fig. 4A and B). We thereby compared the average frequency of specific amino acids in proximity to the crosslinked aspartic- or glutamic-acid in identified CSMs to the theoretically expected distribution. The theoretical distribution was calculated from all, in silico generated, crosslinks that can exist within the acidic library (either crosslinked in separate groups or within one pooled group) under the assumption that every crosslink combination led to exactly one CSM. By that, missing or predominant combinations can be visualized. For both DHSO based datasets (pooled and separate, as shown in Fig. 4C) similar dependencies popped up: Histidine, isoleucine, phenylalanine, tryptophan, tyrosine, and glutamine seem to reproducibly hinder the formation or identification of a crosslink from the acidic library. The frequency of amino acids within identified linear peptide sequence matches of the (non-crosslinked) acidic library was compared to the theoretical amino acid distribution under the

assumption of equimolar peptide quantities (Supplementary Fig. 4C) in an additional experiment. Thereby the same MS method as for crosslink samples was used, meaning that exclusively ions with a charge ≥3 were selected for fragmentation. With this we bias the method to detect longer and higher charged peptides while not recording the majority of linear peptides. This alters the expected amino-acid distribution as peptides with amino acids carrying a positive charge are preferentially detected, enabling a fair comparison to the amino acid-distribution seen in our crosslinked samples. We indeed found fewer peptides containing isoleucine than expected, but clearly more peptides than expected containing histidine. All other amino acids that seemed to have a negative impact on crosslink formation were found in relative frequencies as expected. Except for isoleucine this data strengthens the hypothesis that those amino acids negatively influence the reactivity of DHSO. Especially the basic histidine might cross-react with the activated carboxylic acid to form an intrapeptidal link, therefore impeding the reaction to DHSO.

**Testing the influence of separate FDR calculation and minimal peptide length**. Apart from MeroX, all tested search engines allow their users to decide on performing a separate inter-/ intra-crosslink FDR calculation. MeroX calculates FDR of intra- and inter-protein crosslinks as well as dead-end-links in separate groups by default. A separated target-decoy based FDR calculation is considered useful as the group of interprotein (heteromeric) connections is much larger compared to the theoretical intra-protein crosslinks that can form. This might lead to an underestimated error for the group of heteromeric crosslinks if the FDR is estimated on the total set of CSMs. Lenz et al. showed that by calculating the target-decoy based FDR separately, the final FDR of their DSSO dataset was lowered from 36 to 15%[29]. This is in line with findings from others that found most wrong identifications in the group of interprotein connections especially when using large databases[30,31]. They estimated the error rate to be in the range of 20–25% false positives within a dataset of 2% overall FDR[30]. In contrast to our previously published peptide library[22] consisting of peptides from only one protein, the main library of this study contains 842 theoretical inter-protein crosslinks, 100 intra-protein crosslinks and 100 homeotypic crosslinks (link between peptides of the same sequence). The distribution of inter and intra links nicely represent the theoretical distribution of a real protein mix sample (i.e. E.coli ribosome). In conclusion we were wondering if the FDR calculation in separate groups does also influence our results using the peptide library. In line with our expectations, all tools suffered from a higher error rate within the group of inter crosslinks (Fig. 5A). Interestingly, xiSearch does not show any difference in inter links but a lower FDR for intra links when selecting "ignore groups" (= separate FDR set to off) in xiFDR. Using pLink 2 or MaxLynx the number of crosslinks, but also FDR slightly increases when disabling separate FDR calculation. XlinkX predominantly adds wrong crosslinks to its result file upon enabling separate FDR calculation. Enabling or disabling this option does however not influence the result when using MS Annika. In contrast to our expectations, the separate FDR calculation did not significantly improve overall FDR or ID numbers independently of the search tool used. This might still be reasoned by the nature of our artificial library system that was searched against a database of 171 ribosomal proteins. Hence, peptides of 133 proteins contained in the database are not existent in the sample, leading to a disproportional large number of theoretical vs existing inter-protein crosslinks. Furthermore, the actual number of identified interprotein connections was higher than those of intra-protein

links. This corresponds to the expected theoretical distribution but differs from actual real proteome-wide searches where intra-protein links are more abundant. Aiming to further investigate a more complex system we spiked the peptide library into a non-crosslinked background of tryptic HEK peptides (1:5 mass ratio) and analyzed the resulting data again with or without a separate FDR calculation set in each search algorithm. This however led to a very similar result with little to no effect on the final crosslink IDs. Only with XlinkX we now identify 348 instead of 259 correct interprotein links while maintaining the error rate (Supplementary Fig. 1)

Next, we tested the influence of the peptide length of the shorter peptide within a linked pair on result quality. Figure 5B clearly illustrates that shorter peptides are more prone for wrongly annotated spectra. This fits our expectations as (too) short peptides will generate fewer fragments and therefore yield in less confident identifications. In a large database the chance of sequences from different protein overlapping by chance is furthermore increased with decreased peptide length potentially leading to ambiguous identifications. Based on our data a minimal peptide length of 6 or even 7 seems beneficial, although >100 unique crosslinks are lost when excluding results containing peptides of 6 amino acids length. Of note, our library contains no peptide that has a sequence length of only 5, which is why that group contains exclusively wrong hits.

**Influence of increased sample complexity and crosslink enrichment**. To mimic more realistic conditions—where non-crosslinked linear peptides are way more abundant—we spiked the main library into a tryptic digest of linear HEK peptides at a mass ratio of 1:5. The resulting mixture was analyzed by means of LC-MS/MS and crosslink searches were performed against databases of different sizes, starting with the ribosomal database (171 proteins) that was also used for all other searches and ending with proteome wide searches (Fig. 6A). MS Annika, pLink 2, MaxLynx and xiSearch maintain FDR at levels below 10 % but loose up to 50% or more of their identifications upon increasing the database size to a proteome wide search. Notably, xiSearch maintained high numbers of unique links for all database sizes at quite low FDR rates. XiSearch therefore clearly outperforms all tested search engines by means of ID numbers and FDR in case of proteome-wide searches. On the downside it consumes high computational power, which is why analysis of more than one raw-file at once did not work out for proteome-wide searches on our computer (IntelXenon CPU@ 2.6 GHz, 128 GB RAM). In contrast, MeroX and XlinkX maintain their identification numbers at a high level at cost of data reliability, leading to very high FDR values of up to 26%. The database size furthermore influences the minimal score (maximal for pLink 2) accepted as more decoy hits can be found (e.g. MS Annika increases its accepted minimal score from 120 to 214). As described by Weisbrod and coworkers[16], the number of redundant sequences within the database increases with increasing size leading to ambiguous crosslink IDs. This cannot be visualized with our peptide library system as a correctly annotated crosslinked peptide is still correct in case its sequence is contained several times in the database. Within our CSMs at 1% target-decoy FDR we did not obtain any protein ID that was ambiguous even in case of a proteome wide search. We representatively checked on this issue in MS Annika without any FDR filter and found a maximum of 6.3 % of all CSMs contained at least one redundant sequence. However, in real samples that need biological interpretation such redundant sequences impede proper annotation of CSMs to the respective protein-protein interaction. When enriching the spiked library by size exclusion chromatography (SEC) we were able to (re-)boost

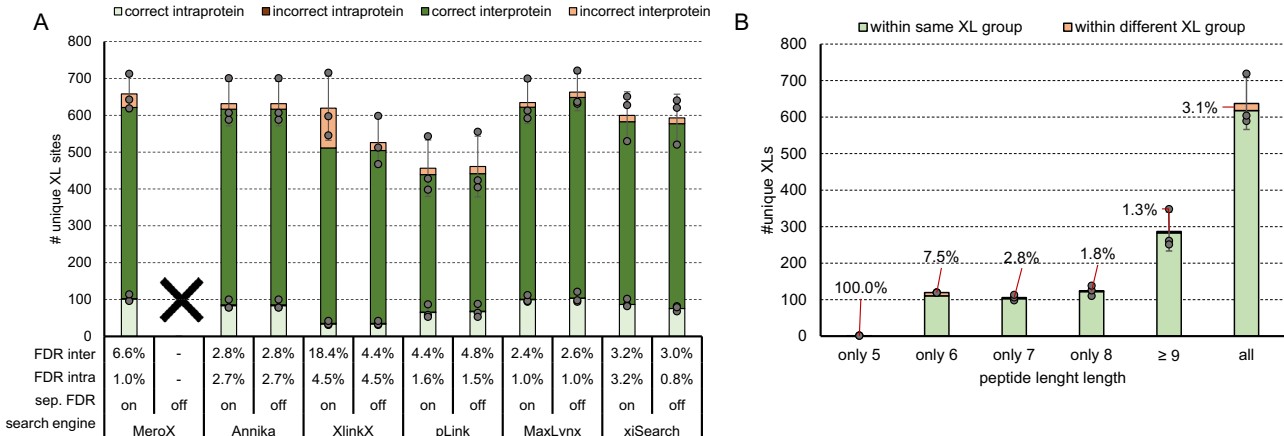

**Fig. 5 Effect of separate inter/intra FDR calculation and minimal peptide length on FDR. A** Average number of unique crosslinks from the DSSO crosslinked main library after acquisition using a stepped HCD MS2 strategy with separate FDR calculation for inter- and intra-crosslinked peptides set on or off. Although synthetic peptides were used for crosslinking their sequences are based on ribosomal protein sequences. "Intraprotein" are defined as homomeric links and "interprotein" are heteromeric links based on the proteins the synthetic peptides correspond to. **B** Average number of crosslinks from the DSSO linked main library identified with MS Annika at 1% estimated FDR when filtering for crosslinked peptides of the given length (meaning the length of the shorter peptide within the crosslinked sequence). **A** and **B** Error bars indicate standard deviations of average values, experimentally validated FDR is shown as callout or in table format under the x-axis, $n = 3$ independent samples acquired on different days. Source data are provided as a Source Data file.

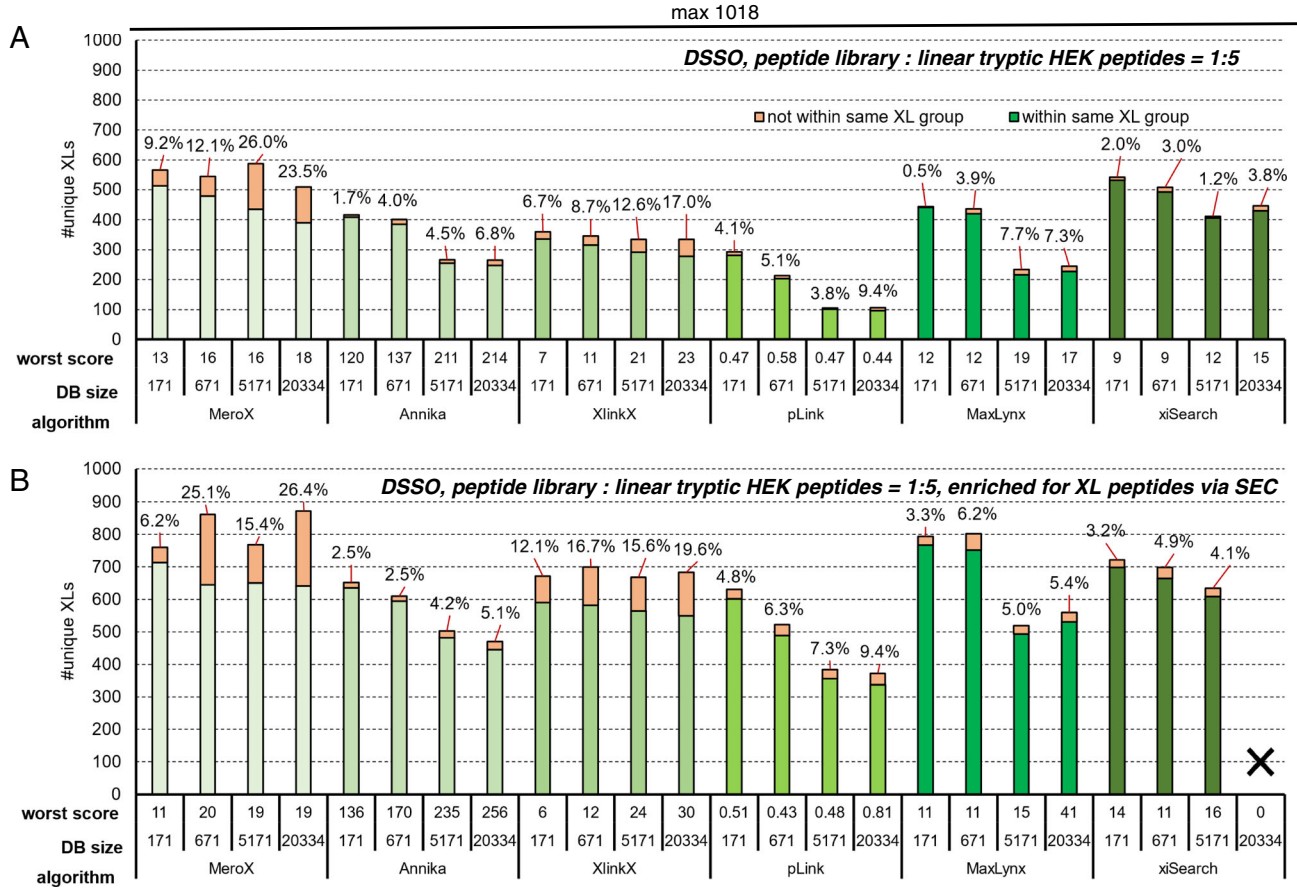

**Fig. 6 Performance benchmarking in a mimicked complex environment and upon increased database size.** The DSSO linked main library was mixed with linear tryptic HEK peptides (1:5 w/w). Bars indicate the number of unique crosslinks after acquisition using a stepped HCD MS2 strategy and identified using the indicated algorithm at 1% estimated FDR when using databases containing exclusively 171 E. coli ribosomal proteins, or additional 500, 5000, or 20163 human proteins. **A** direct measurement (**B**) measurement after enrichment for crosslinked peptides by size exclusion chromatography. Of note, analysis of the 5 SEC fractions did reproducibly not work with our largest 20334 protein database and xiSearch, as the software crashes. This data is therefore missing. Source data are provided as a Source Data file.

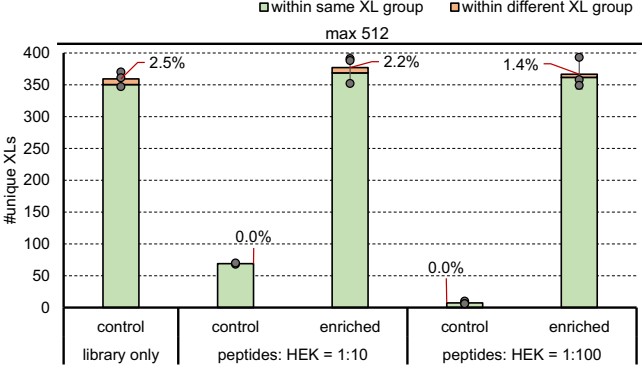

**Fig. 7 Affinity enrichment of DSBSO crosslinked synthetic peptides from a complex environment.** Average number of unique crosslinks after acquisition using a stepped HCD MS2 strategy identified in the DSBSO crosslinked enrichable library with or without spiking using linear tryptic HEK peptides as indicated. Controls were directly used for measurement; enriched samples were subjected to affinity enrichment. Error bars indicate standard deviations of average values, experimentally validated FDR is shown as callout, $n = 3$ independent samples acquired on different days. Source data are provided as a Source Data file.

identifications to the level seen without spiking (Fig. 6B) and obtaining the same trends with regards to FDR. We additionally applied post-score cutoffs to the results using the largest database based on the scores that yielded in 1 % experimentally validated FDR in our initial non-spiked measurements (shown in Fig. 2A) and that are more stringent than those cutoff values recommended by the authors of the respective search engines. This improves the experimentally validated FDR, that is however still ranging from 2.1% for pLink 2 to 10.5% for MeroX in the spiked samples (Supplementary Fig. 2A and B). The database size dependent effects we observed within our spiked samples are furthermore reproducible when analyzing the non-spiked library with the same set of databases, as representatively analyzed with MS Annika and shown in Supplementary Fig. 2C. Our results suggest that the choice of a properly sized database is of high importance for the reliability of the results as well as that post-score cutoffs to minimize effects of improper FDR estimation need to be empirically determined in dependence of used software and complexity of the sample.

To check for the performance of affinity-enrichment using the azide tagged linker DSBSO we also spiked the enrichable library, containing no azide-protected lysines, with linear tryptic HEK peptides in mass ratios of 1:10 or 1:100 (Fig. 7). Enrichment was performed by clicking crosslinked peptides to beads functionalized with dibenzocyclooctyne (DBCO) as previously described[32].

The total amount of peptides subjected to MS analysis was kept constant at 1 μg for all injections as this seemed maximal for our LC-MS setting. This means that the 1:100 control sample contains 10 ng crosslinked peptides. For enrichment 20 μg crosslinked peptides were spiked with 200 μg or 2 mg HEK peptides resulting in 1.3 μg total peptides on average, and independent of the spike ratio, in the enriched fraction. Although quite some input material was lost during enrichment, the theoretical input can be freely upscaled to compensate. The resulting enriched samples were of high purity, enabling the injection of close to 1 ug crosslinked material even in samples with high amounts of background (instead of only 10 ng, as in the 1:100 spiked control) and therefore maintaining constantly high crosslink numbers and low FDR values independently of the sample complexity prior to enrichment.

**Influence of additional FAIMS separation on resulting crosslink identification numbers and properties.** High-field asymmetric-waveform ion-mobility spectrometry (FAIMS) adds another separation dimension and therefore decreases spectrum complexity and reduces noise. Both effects were reported beneficial for the identification of crosslinked peptides[20]. We probed the effect of FAIMS on the here presented synthetic peptide system using DSSO (Supplementary Fig. 3). In line with the results of Schnirch and coworkers[20], we observe a maximum number of unique crosslinks when using compensation voltages (CV) in the range of −50 to −60V. Furthermore, we observed very high reproducibility when using FAIMS and a trend to lower FDR values upon lowered CV. The combination of 3 CVs within one run boosted our overall identification number to 700 which is a 10 % increase compared to our measurements without FAIMS (numbers from analysis using MS Annika at 1 % estimated FDR).

Next, we investigated if lower injection amounts are sufficient for identification of crosslinks thanks to the improved signal to noise ratios when using FAIMS[33,34]. Without FAIMS, on the HFX 1000 ng were injected for all samples, which was stepwise lowered down to 10 ng on the Exploris with and without FAIMS. Judged from ID numbers, 100 ng, only 1/8 of the maximal amount used, was sufficient to still identify close to 500 unique crosslinks with FAIMS which is ~90% of the identified links using 800 ng input (Fig. 8A). Further lowered amounts lead to a drastic decrease of crosslinks. In a direct comparison of data acquired with/without the FAIMS device attached, we see a clear advantage of FAIMS especially for lowered injection amounts based on the number of identified crosslinks. Upon injection of higher peptide input the effect seems to diminish but the number of CSMs found per unique crosslink is still increased by using FAIMS (Supplementary Fig. 8A).

Since our data with FAIMS outperforms those without we focused further investigations on FAIMS data: When comparing retention times of crosslinked peptides IDs, we observed a slight shift towards higher retention times with higher peptide amounts (Fig. 8B). This effect is not reasoned by an overloaded column as the retention time of individual CSMs did not change (Supplementary Fig. 8B), but rather by identifying additional CSMs. When looking into the physicochemical properties of the additionally identified CSMs, a shift in hydrophobicity (Fig. 8C) as well as an increase of molecular weights and m/z (Fig. 8D) can be observed, which might explain shifted retention times upon increasing input amounts. Furthermore, the relative charge distribution (Fig. 8E) depends on the input amount. The fraction of high charged $z = 5$ ions is increased, while the fraction of $z = 4$ charged ions is lowered with higher input. This observation is in line with the seen dependency of molecular weight to input amount as larger peptides are more likely highly charged in an acidic environment. When comparing this charge distribution to those observed without FAIMS (Fig. 8F) a clear shift from more dominant $z = 3$ charged ions without FAIMS to predominant $z = 4$ charged ions with FAIMS can be observed, which is beneficial for the detection of predominantly higher charged crosslinked ions over linear peptides. Of note, this effect seems to be dependent on the used instrument type as well, since the relative fraction of +3 charged CSMs is the biggest in our data from the HFX. Those results without FAIMS obtained from 25 to 50 ng input contain only a total of 6 and 24 CSMs respectively, which is why the relative results seem not to fit to those results with higher ID numbers. With only 10 ng of input no CSMs were identified at 1% FDR (Supplementary Fig. 8A).

**Benchmarking acquisition strategies.** We finally investigated the FDR of crosslinks using different MSn acquisition strategies on an

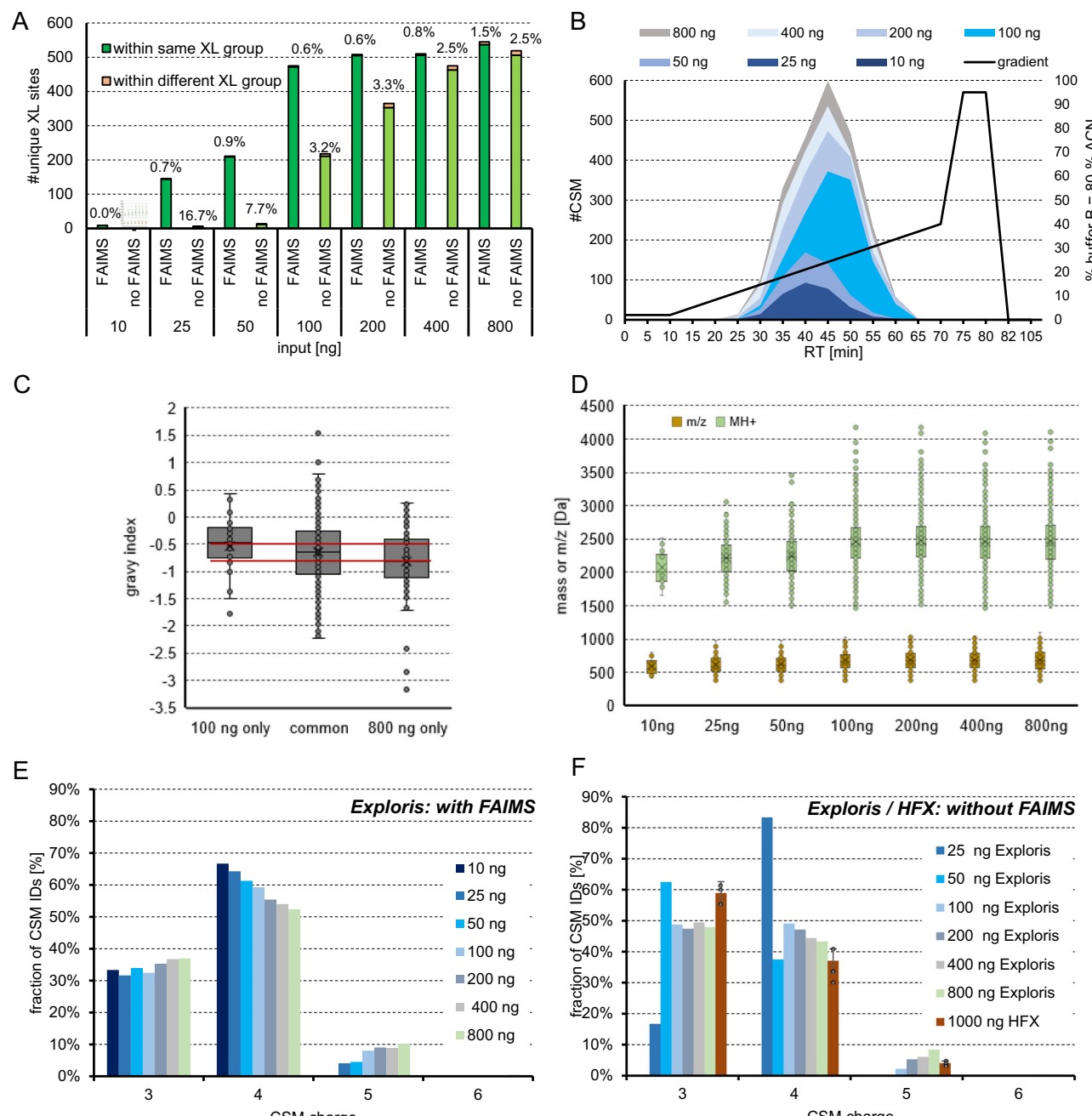

**Fig. 8 Variation of input amount and physicochemical properties of crosslinks.** Lowered input amounts, as indicated, of the DSSO cross-linked main library were measured on an Orbitrap Exploris 480 using a stepped HCD MS2 strategy. Data was analyzed using MS Annika at 1 % FDR. **A** Unique crosslinks with or without FAIMS attached and experimentally validated FDR is given above bars. **B** Distribution of spectral matches over retention time and used gradient with FAIMS. **C** Distribution of hydrophobicity of CSMs identified exclusively in the 100 or 800 ng sample or in both samples (measured with FAIMS). **D** Distribution of identified cross-linked peptide masses of CSMs (M + H) and m/z in dependence of used input amount. (measured with FAIMS) **C**, **D** Boxplots depict the median (middle line), upper and lower quartiles (boxes), 1.5 times of the interquartile range (whiskers) as well as outliers (single points), this data is from a single replicate. **E** Charge distribution obtained with FAIMS using CVs −50, −55 and −60 for acquisition and stepwise changing the input amount from 10 – 800 ng or (**F**) without FAIMS when acquiring on HFX or Exploris instrument. Error bars indicate standard deviations of average values, experimentally validated FDR is shown as callout, $n = 3$ independent samples acquired on different days. Source data are provided as a Source Data file.

Eclipse Tribrid mass spectrometer (Thermo Fisher Scientific). Data analysis was performed using XlinkX at 1 % FDR on CSM and residue pair level. MS3[35] methods are reported as more reliable for crosslink identification when using cleavable cross-linker. Therefore, MS3 (acquired as described in[35]) was compared to an MS2-MS2 method (acquired as described in[22]) and our standard stepped HCD acquisition method for the main library

crosslinked using DSSO (Fig. 9). In line with previous results from our group[17,22], stepped HCD outperformed MS3 and MS2-MS2 results in terms of unique crosslink numbers. We however observed a slight increase in FDR (8.5% vs 7.6 and 7.2% for MS3 and MS2-MS2 respectively). Surprisingly, the experimentally validated FDR for MS3 is higher with lower unique residue pair numbers than for the MS2-MS2 method which contrasts with the

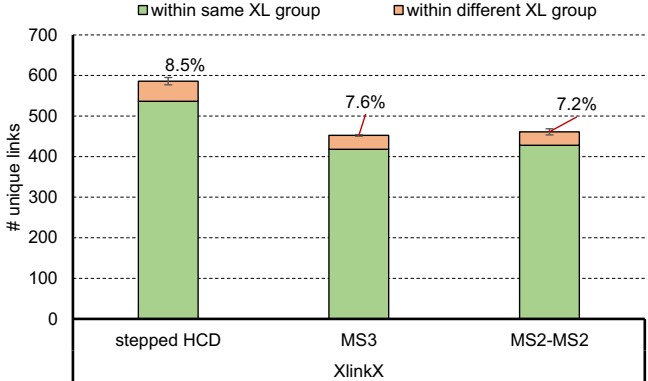

**Fig. 9 Comparison of experimentally validated FDR of MSn methods.** DSSO cross-linked main library were measured on an Eclipse Tribrid Mass Spectrometer using a stepped HCD MS2, MS3 and CID MS2- ETD MS2 acquisition strategy. Data was analyzed using XlinkX at 1% FDR. Red bars indicate false crosslinks whereas green bars represent true unique crosslinks. Error bars indicate standard deviations of average values, experimentally validated FDR is shown as callout, $n = 3$. Source data are provided as a Source Data file.

literature. Although MS3 and MS2-MS2 methods are thought to give advantage for crosslink identification, stepped HCD performed better in our hands.

## Discussion

The here presented peptide libraries represent a valuable and highly flexible standard to benchmark crosslinker reagents of different chemical reactivity as well as for comparison of search engines or acquisition strategies. Thanks to peptide sequences originating from 38 different proteins, the library represents a realistic digest from the *E. coli* ribosomal protein complex, allowing for in-depth analysis of search-engine specific FDR calculation. Our in house developed tool IMP-X-FDR comes with an easy-to-use user interface and allows FDR calculation, comparison of crosslink results across software platforms and investigation of crosslink-properties in an automated manner also for non-bioinformaticians.

Our results suggest that additional and empirical score cutoffs are a valuable instrument to correct the actual FDR. The height of this cutoff value is not only software specific but also dependent on database size and sample complexity. Our library helps to find such specific score-cut-off values but also showed that built-in target-decoy based FDR estimation overestimates correct results in case of all tested search engines. Of note, not all tested tools allow to perform their target-decoy based FDR estimation on unique crosslink or even protein-protein interaction level but instead on spectrum (CSM) level: To give an example, MeroX and pLink 2 do their FDR estimation on spectrum level, leading to false hits propagating during grouping to the final unique crosslinks and higher than expected final experimentally validated FDR[36]. We believe, machine learning approaches as well as the inclusion of additional parameters like retention time or ion mobility to (re-)score identifications might improve crosslink numbers and validity of FDR in future algorithms.

The most challenging part of data analysis seems to find a good compromise in the trade between high identification numbers and low FDR. In our data, MS Annika and MaxLynx seem to find this optimum best for cleavable crosslinkers, while pLink 2 seems to perform very well for the tested non-cleavable linker. The performance of xiSearch was stable for non-cleavable and cleavable crosslinkers as well as for database sizes up to 5000 proteins. In conclusion, xiSearch can be seen as an allrounder within the

crosslink search engines compatible with both crosslinker types. We however faced troubles in software stability for proteome-wide searches. We further observed that some search engines fit better to specific linker reagents than others leading to an additional linker dependent performance difference that is not caused by the crosslinker chemistry itself but by technical reasons as an altered spectrum complexity for MS cleavable vs non-cleavable reagents. This is even reflected in issues to properly define specific new linker reagents. To give an example, sulfoxy based linkers as DSSO or DSBSO were reported to generate characteristic doublet peaks of two different delta masses upon MS fragmentation thanks to water elimination[12,37]. Upon the tested algorithms, MaxLynx and pLink 2 do not allow a definition of more than two fragments and their results might be further improved upon implementation. Another example is Thermos Proteome Discoverer 2.5, were the definition of a fragment mass 0 is impossible but needed for the zero-length linker CDI. As a workaround a very low mass $\geq 1\text{E}^{-5}$ must be defined. This affects search engines as XlinkX or MS Annika when running as a node within Proteome Discoverer. In line with these observations, it seems that the developers of search engines focus on specific linker types for optimization of their algorithms and this yields in boosted results and better score-based separation of target vs decoy hits for linkers of the exact same chemistry.

Many studies aim to minimize error rate and maximize confidence in crosslink results with alternative approaches: The Rappsilber laboratory investigated this issue by separately crosslinking fractions after size exclusion chromatography (SEC) and accepting only those protein-protein interactions as confident that are between proteins of the same SEC fraction[29]. Another common way to validate crosslink data is by comparison to 3D structures of representative protein complexes in the dataset. Yugandhar et al.[38] showed that this approach can lead to a significant underestimation of the actual error rate by implementing additional quality criteria, including the validation of interactions by orthogonal techniques, by known interactions or by adding the proteome of unrelated organisms to the search and checking for misidentifications. In line with our results their results further show, that applying minimal score-cutoffs can drastically reduce error rate and might therefore be highly beneficial to obtain interpretable and confident results. The Bruce lab estimates the error rate for large scale studies by determining the theoretical maximal number of inter- and intra-protein crosslinks based on available 3D structures. They demonstrate that those inter-protein crosslink fractions greater than the theoretical maximum value are most likely occurring from false positive IDs[39].

Complementing these studies, a synthetic library system serves as ground truth model to experimentally validate observed FDR. We believe that a gold standard in the field of crosslinking MS must be established in the future for robust data analysis. Further software updates or novel algorithms will improve the reliability of the results and increase the coverage of crosslink identification.

Our data will therefore provide valuable input to benchmark new or updated search engines. The freely available IMP-X-FDR can be easily adopted for automated FDR calculation with any novel crosslink search engine thanks to the open-source code. Furthermore, improvements in crosslinker reagents, MS instrumentation or chromatography can be validated using the physical library where the exact number of theoretically reachable crosslinks is well defined.

## Methods

**Peptide synthesis.** Solid phase peptide synthesis was done using Fmoc chemistry on a SYRO with Tip Synthesis Module (MultiSynTech GmbH). Each coupling step was performed as double coupling using HATU/DIEA for carboxylic acid activation. Lysine residues at the C-terminus bore an azide group instead of an amine to

hamper any cross-linking at this position. N-termini were designed as acetyl protected WGGGGR sequence tag and C-termini were designed as amide protected RGGGG sequence tag (for peptides to be used with linkers reactive to acids, see Supplementary Data 1 for all sequences). For this Fmoc-L-Arg(Pbf)-TCP (# PC-01-0126), Fmoc-Rink-Amide-(aminomethyl) (#PC-01-0501) or Fmoc-L-Lys(N3)-TCP (custom synthesized) resins were used respectively (all: INTAVIS Peptide Services GmbH & Co. KG). Purification was performed using a C18 kinetex column (5 μm) and a 30 min gradient. All peptides were analyzed using a 4800 MALDI TOF/TOF (Applied Biosystems) for quality control purposes. Lyophilized peptides were solubilized in water and their concentration was estimated by measuring their absorption via a nanodrop (DeNovix DS-11 FX +) at 280 nm and calculating the sequence specific extinction coefficient using the ProtParam tool[40]. Peptide solutions were dried under reduced pressure, resolubilized in 50 mM HEPES pH 7.5 at a concentration of 5 mM and mixed to groups for cross-linking (Supplementary Data 1).

**Sample preparation**. For lysine reactive cross-linker reagents (DSSO, DSBSO, DSBU, CDI) 9.3 mM cross-linker reagent stock solutions were freshly prepared in dry DMSO. 0.5 μL of stock solution was added to 1 μL of each peptide group in separate vials. Additional stock solution was added 4x every 30 min adding up to a total of 2.5 μL cross-linker stock solution. The resulting 3.5 μL reaction mix were quenched using 31.5 μL 100 mM ammoniumbicarbonate (ABC) buffer for 30 min and pooled to a single tube. The resulting mix was digested by addition of 5 ng trypsin/group over night at 37 °C. Azide protection groups were finally reduced to the respective amines by incubation to 50 mM (final concentration) tris(2-car-boxyethyl)phosphine (TECEP) for 30 min at room temperature. Reduced peptides were pooled to a single vial, aliquoted and stored at −70 °C upon further usage.

For aspartic acid and glutamic acid reactive cross-linker reagents (DHSO, ADH) 300 mM cross-linker reagent and 1.2 M (4-(4,6-dimethoxy-1,3,5-triazin-2-yl)−4-methyl-morpholinium chloride) (DMTMM) stock solutions were prepared in 25 mM HEPES pH 7.5. 0.25 μL of cross-linker and DMTMM stock-solution were added 5x every 30 min to 1 μL of each peptide group. The reaction was quenched by adding trifluoracetic acid (TFA) to a final concentration of 4 % (w/v) for 20 min followed by re-neutralization by addition of 50 μL 1 M Tris pH 7.5 buffer. Peptides were pooled and digested as described above.

**Enrichment strategies**. To mimic complex mixtures, cross-linked and digested peptide pools were mixed with a 5-100x excess (by mass) of tryptic HEK peptides. The resulting spiked samples were enriched either by size exclusion chromatography (SEC) or via affinity enrichment.

For SEC, ~10 μg of cross-linked peptide-library + typtic HEK peptides were fractionated on a TSKgel SuperSW2000 column (300 mm × 4.5 mm × 4 μm, Tosoh Bioscience), which was operated at 200 μl/min in 30 % ACN, 0.1 % TFA. Fractions were collected every minute, ACN was removed under reduced pressure to obtain a concentrated sample for LC-MS/MS

DSBSO cross-linked peptides (+ linear HEK peptides in varying mass ratios) were affinity enriched using dibenzylcyclooctyne (DBCO) immobilized to beads as described elsewhere[32]. Briefly, NHS-activated Sepharose fast flow (#17-0906-01, GE Healthcare) an an 2x molar excess of DBCO-amine (#761540, Sigma-Aldrich) for 1 h at room temperature followed by 6 x washing steps using 50 mM HEPES pH 7.5. A 10x molar excess (DBCO groups to DSBSO linker) of DBCO coupled Sepharose beads was added to the sample and incubated for 2 h at room temperature under continuous rotation. Beads were washed with 5 bead volumes, 3x using 50 mM HEPES pH 7.5 with 1 M NaCl, 3x with 10 % ACN in H2O and 3x with 10 mM Tris pH 7.5. To elute crosslinked peptides, beads were incubated to one bead volume of 2% (v/v) trifluoracetic acid in H2O for 1 h at room temperature.

Tryptic HEK peptides were generated as follows: HEK293T Lenti-X (TaKaRa bio, Cat# 632180) cells were lysed in 10 M urea in 100 mM Tris by ultrasonication. The cleared lysate was reduced at a final concentration of 10 mM dithiothreitol in the presence of benzonase for 1 h at 37 °C. This was followed by alkylation at a final concentration of 20 mM iodoacetamide for 30 min at room temperature in the dark. Digestion was performed using LysC (1:200 w/w) for 2 h at 37 °C in 6 M urea followed by addition of trypsin (1:200 w/w) for 16 h t 37 °C in 2.5 M urea.

**Chromatographic separation and mass spectrometry**. Samples were separated using a Dionex UltiMate 3000 HPLC RSLC nano-system coupled to a Q Exactive™ HF-X Orbitrap mass spectrometer or to an Orbitrap Exploris™ 480 mass spectrometer equipped with a FAIMS pro interface (all: Thermo Fisher Scientific). Mass spectrometers used for data acquisition were operated using Thermo Scientific Xcalibur v4.2.4.7 (HF-X devices) or v 4.4.16.14 (Exploris devices). Samples were loaded onto a trap column (Thermo Fisher Scientific, PepMap C18, 5 mm × 300 μm ID, 5 μm particles, 100 Å pore size) at a flow rate of 25 μL min-1 using 0.1 % TFA as mobile phase. After 10 min, the trap column was switched in line with the analytical column (Thermo Fisher Scientific, PepMap C18, 500 mm × 75 μm ID, 2 μm, 100 Å). Peptides were eluted using a flow rate of 230 nl min−1, with the following gradient: 0–10 min 2 % buffer B, followed by an increasing concentration of buffer B up to 40% until min 130. This is followed by a 5 min gradient from reaching 95 % B, washing for 5 min with 95% B, followed by re-equilibration of the column in buffer A at 30 °C (buffer B: 80 % ACN, 19.92 % H2O and 0.08 % TFA, buffer A: 99.9% H2O, 0.1% TFA).

The mass spectrometer was operated in a data-dependent mode, using a full scan (m/z range 375-1500, nominal resolution of 120.000, target value 1E6). MS/MS spectra were acquired by stepped HCD using an NCE (normalized collision energy) of 27 ± 6 for sulfoxy group linkers (DSSO, DSBSO, DHSO), 30 ± 3 for urea-based linkers (DSBU, CDI) and 28 ± 4 for non-cleavable linkers (ADH). An isolation width of 1.0 m/z, a resolution of 30.000 and a target value of 5E4 (on HF-X) and 1E5 (on Exploris) was set. Precursor ions selected for fragmentation (±10 ppm, including exclusively charge states 3-8) were put on a dynamic exclusion list for 30 s. Measurements including FAIMS were performed on the Orbitrap Exploris under alteration of used compensation voltages as given for each result.

MS3 and MS2-MS2 acquisitions were performed on a Orbitrap Eclipse Tribrid (Thermo) using the same HPLC setting as described above. Acquisition strategies were designed as described in Wheat et al.[35] and Beveridge et al.[22] respectively.

**Data analysis and post processing**. Data analysis was performed against a custom shotgun database containing 171 E. coli ribosomal proteins at 1 % FDR level. For analyses using MS Annika or XlinkX, Thermo raw files were loaded to Thermos Proteome Discoverer 2.5 that and both search engines were used as node within that software. MaxLynx was used as part of MaxQuant v 2.0.2.0 by direct usage of Thermo raw files as well. For MeroX, raw files were converted to mzML and for pLink 2 and xiSeach files were converted to mgf using MSConvertGUI v3.0.21084. The result files are available for download in the PRIDE repository[41] using the identifier PXD029252. The software specific settings are furthermore summarized in Supplementary Data 3.

Post processing was done using the graphical user interface of our in house developed tool IMP-X-FDR (Supplementary Fig. 5E). It enables to calculate the experimentally validated FDR and therefore validate the target-decoy based FDR estimated by the crosslink search engine according to the following formulae:

$$FDR_{experimentally\ validated} = \frac{target\ XLs\ across\ peptides\ not\ within\ same\ XL\ group}{target\ XLs\ total}$$

When calculation FDR on CSM level, unique residue pairs (XLs) are replaced by CSM IDs in the above formulae. Some search engines allow the export of target-decoy filtered XL lists, but not all of them. To ensure functionality with all search engines and enable the direct usage of the search engine result file as input for IMP-X-FDR, our tool automatically filters away IDs marked as decoy and exclusively selects inter- and intra-protein crosslinks (but excludes dead-end links or linear peptides).

FDR validation is done based on crosslinks only allowed as correct in case they are formed within the same crosslink group (see Supplementary Data 1 for allocation of peptides to groups). We call this functionality "FDR recalculation" and adopted the code for each crosslink search engine, due to differences in their output format. For a correct FDR recalculation, a support file containing all group-allocated peptides of all used (sub) peptide libraries is provided with the software. The tool outputs a csv file containing a list of al XLs within the same or different group as well as informative graphs showing the number of IDs and the score vs experimentally validated FDR or number of crosslinks (Supplementary Fig. 5A–C). The functionality "Venn diagrams" of IMP-X-FDR was used to visualize the overlap of replicates of searches from different search algorithms (example output shown in Supplementary Fig. 5D). This functionality uses the output of "FDR recalculation" as input, which ensures a uniform format and compares peptide sequences, their originating protein, and the position of the peptide in that protein.

The third function of IMP-X-FDR is to investigate physicochemical properties of crosslinks. To do so the freely available tools from Biopython 1.79[42], specifically from the Bio package, Bio.SeqUtils subpackage and Bio.SeqUtils.ProtParam module, were used. Furthermore IMP-X-FDR was operated using the following modules: Matplotlib_venn 0.11.6, openpyxl 3.0.9, toolz 0.11.2, venn 0.1.3, xlrd 1.2.0, XlsxWriter 3.0.3. Crosslinked peptides were represented in a linearized form to ensure compatibility with the used packages originally designed for linear peptides. IMP-X-FDR outputs a csv file containing calculated crosslink properties, which includes the isoelectric point, fraction of aromatic amino acids, molecular mass, gravy value and amino acid distribution. The obtained data is automatically compared to the respective properties of all (in silico generated) theoretically formed crosslinks within the library. Thereby we assume the identification of exactly one CSM for each theoretical crosslink. The unnormalized output graphics are constructed on the crosslink level and histograms constructed on CSM level are normalized to a total area of 1. Finally IMP-X-FDR investigates amnio acid motives using the module seqlogo 5.29.8[43] to create position probability matrices. Thereby the closest three neighboring amino acids of the linker's binding site are investigated for frequent or rare amino acids and can be compared to the (theoretically expected) crosslinks within the library. Representative output graphs are illustrated in Supplementary Fig. 6. A user's manual, containing a detailed explanation of each output file and used functions is delivered with IMP-X-FDR. The code is freely available (https://github.com/fstanek/imp-x-fdr[23]) and can be used on command line basis or via a graphical user interface.

**Reporting summary**. Further information on research design is available in the Nature Research Reporting Summary linked to this article.

## Data availability

All raw and result files are available for download in the PRIDE repository[41] using the identifier PXD029252. Source data are provided with this paper.

## Code availability

The code of IMP-X-FDR available from Github (https://github.com/fstanek/imp-x-fdr) and Zenodo[23].

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

## Acknowledgements

This work supported by the following third-party funding granted to K.M.: This work supported by the EPIC-XS, Project Number 823839, funded by the Horizon 2020 Program of the European Union, by the project LS20-079 of the Vienna Science and Technology Fund and the by the ERA-CAPS I 3686 and P35045-B project of the Austrian Science Fund. We thank the IMP for general funding and access to infrastructure and especially the technicians of the protein chemistry facility for continuous laboratory support. Our gratitude further goes to Dr. Elisabeth Roitinger for fruitful discussions and for her valuable inputs to the manuscript.

## Author contributions

The study was designed by M.M. and K.M. Experiments were performed by ADV. IMP-X-FDR was created by ADV and wrapped into a user interface by F.S. M. Madalinski performed peptide synthesis. Experiments were performed and the manuscript was written by M.M. F.M. added data and figures for results of xiSearch and MS3, MS2-MS2 acquisition strategies and helped with the revised version of the manuscript. All authors have given approval to the final version of the manuscript.

## Competing interests

The authors declare no competing interests.

## Additional information

**Peer review information** : *Nature Communications* thanks Michael Trnka, and the other anonymous reviewer(s) for their contribution to the peer review of this work. Peer reviewer reports are available.

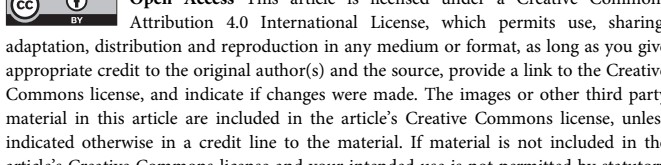

