## [Peer Review File · Nature Communications]

REVIEWER COMMENTS

Reviewer #1 (Remarks to the Author):

The manuscripts from the Mechtler group expands upon a previously published library of crosslinked peptides intended for benchmarking and optimization of CLMS analytic strategies. Namely it provides a means of estimating the true FDR of an analytic platform and enables the evaluation of the platform's ability to accurately assess FDR and a means to compare the performance of different strategies against the same benchmark. The main differences from the previous publication (Beveridge et al ref 22), is that this library is larger (1018 theoretical crosslinks vs 426 in previous version) and they have included a library suitable for benchmarking reagents that target acidic residues. An additional library (or really just a subset of the main one) is compatible with azide-containing crosslinker, commonly used in click based affinity enrichments. Finally, they also include a software tool that calculates FDR as well as physiochemical properties of crosslinked peptides to compare against theoretical distributions as a means to check the reliability of results.

I work in this field and have found the previous library to be an extremely valuable resource for developing and testing our FDR estimation software, and this newer library will no doubt continue to be a useful resource to people working in the field. I question the novelty of this particular paper, since it is just a minor expansion of the same principle as before. However, I will let the editor decide if the novelty is sufficiently to meet Nature Communications standards.

The main downsides I see to this work are that the library only focuses on cleavable crosslinkers acquired using MS2 based strategies. There have been some recent publications (for instance Yugandhar, MCP 2019) that have tried to do similar work and found that MS2 only strategies were far less reliable than those incorporating MS3. Hence, it would be desirable to have the same library acquired with MS3 based acquisition instead of just using MS2 as done here. Even, if the FDR can be controlled by MS2 strategies, as shown here with the larger DB searches containing HEK spike-ins, it would be great to compare the performance and trade-offs of MS3 based approaches by acquiring the library using MS3, or at least, to enable others to do so. Additionally, it is a bit disappointing that they do not include results for BS3 acquired using a simple fixed HCD energy as this is still an incredibly useful and common approach. However, this is less of an issue since people developing these search engines can still use the previous library, and the MS2 results should still be applicable here (for instance pLink is tested).

An additional omission is that there is no discussion of site-localization or benchmarking site-localization strategies. For Lys directed CLMS this is not typically an issue since essentially all

peptides have a single modifiable Lysine residue, but it is a problem for the acidic-CLMS library. The strategy here seems to be to skirt the issue by creating peptides with only one possible site of modification, but this isn't a realistic scenario as most peptides will have multiple D/E residues.

Minor-

Figure 1:

Clearly label or caption the figure to make clear that data were acquired by MS2 centric strategies.

They test pLink2, but the text refers to pLink. I would use pLink2 in the main test, at least at the first mention, for clarity. It is interesting that pLink2 performed best in the previous BS3 benchmarking study. This is worth discussing that pLink2 is using an open-modification strategy, while the other software are pre-determining the individual peptide masses based on the diagnostic ions, so quite a different algorithm. Also, statements in the text that pLink wasn't optimized for HCD don't make sense (pg 101)... pLink is primarily used with HCD data.

In panels B/C they show overlap in unique XL ids between different experimental replicates searched with Annika (B) and the same replicate searched with different search engines (C). They show in C) that FDR is low for the area of overlap compared to XLinks IDed by just one search engine. I would like to know if this is also true in panel B).. how much better is FDR in XLs identified twice, and is it any better to use two separate searches of one replicate rather than one algorithm on two replicates?

Figure 2:

I would find it helpful to have chemical structures in the figure so that I don't have to look them up. For instance, I didn't understand why the enrichable library was being tested until I realized that DSBSO is an azide containing reagent. The caption is mislabeled at one point B/C are incorrectly referenced.

Figure 3:

Seems to be missing a caption for panel C. The results in the supplemental Figure 3 exploring the effect of neighboring amino acids on acidic crosslinking are interesting. I would consider using a standard Logo plot in Sup Figure 3A, as I found their data visualization a little hard to understand. Why does it seem like Leucine enhances crosslinking while Isoleucine inhibits?

In line 199 of the text, I don't understand what they are trying to say with respect to selection of 3+ and higher precursors.

Figure 4:

I'm surprised there is not much effect from setting separate intra-protein and inter-protein crosslink thresholds. I typically see many more intra-protein hits at the expense of fewer inter-protein hits. It would be useful if they reported stratified results like this. Also, the magnitude of this effect could depend a lot on the size of the data base being searched. I think it would be more informative to perform this analysis on the HEK spike in sample.

Figure 5/6 are nice.

Figure 7:

I am surprised they mostly see 3+ precursors with DSSO -FAIMS. Is this reporting the charge of the CSMs (as indicated in the y-axis label) or all spectral matches (including linear peptides?). Figure S2 would be improved by addition of no-FAIMS data, but apparently they only collect this on the HFX and not the Exploris? It's not that hard to take off the FAIMS interface.

Reviewer #2 (Remarks to the Author):

In this work, Matzinger et al presented an improved version of the synthetic peptide library for cross-linking mass spectrometry (XL-MS) studies (previously developed and published by their lab – Beveridge et al., Nature Communications, 2020). As with any high-throughput studies, quality-control is one of the most important aspects of XL-MS. Specifically over the past couple of years,

some key papers have been published, demonstrating the existing quality-control issues in XL-MS along with potential solutions to address those (including the synthetic peptide library from the authors' own lab). Thus, establishing methods to carefully examine the quality of the results (including the development of the synthetic peptide library by the authors) is important for the rapidly evolving XL-MS field. However, the current manuscript is mainly an improvement on their previous synthetic peptide library (Beveridge et al. 2020) and might not be significant enough for a new publication especially in a journal of the high caliber as Nature Communications.

Other specific comments:

1. As for benchmarking different acquisition strategies, I could not find any analysis for additional strategies such as MS2-MS2 and MS2-MS3 (with different types of energies; similar to the Figure 7c from their original benchmarking paper Beveridge et al., Nature Communications, 2020).
2. Does their IMP-X-FDR tool allow 'FDR recalculation' at different levels (such as CSMs, residue pairs and protein pairs), similar to how the Xi FDR tool from Rappsilber lab allows for FDR estimation at different levels? If not, it would be a good functionality to add.
3. The authors should compare the utility of their synthetic peptide library with that of existing quality assessment metrics for XL-MS, such as (i) the metric from Keller et al (Journal of Proteome Research, 2019) that accounts for the fraction of interprotein XLs and (ii) metrics from Yugandhar et al (Nature Methods, 2020) that accounts for corrected structure-based validation of XLs, along with metric utilizing additional false search spaces.
4. Adding a discussion on why different search tools perform better for specific linkers and specific acquisition strategies (software-specific results) would be informative. For example, their MS-Annika seems to be performing better on DSSO than DSBU (as described in the paper as well; Figure 2). Also, MeroX seems to perform drastically better on DHSO samples in comparison to ADH samples in terms of both sensitivity and specificity. Additionally, it is very interesting to note that in Figure 4A, XlinkX seems to perform drastically poorer when separate FDR calculation was turned on, which is not expected considering the advantage the separate FDR calculation has on the quality-control. I can understand that different search tools might be designed differently. However, drawing from the authors' group's own experience with development of cross-link search software, a thorough discussion relating the key technical aspects of cross-link identification with such soft-ware specific results would add value for this work.
5. Minor: for efficient reproducibility of the results, please provide full protein sequences that were utilized as search databases for different analyses performed in this study in supplementary information.

Reviewer #3 (Remarks to the Author):

The authors present a set of tools and software to benchmark analytical pipelines for cross-linking mass spectrometry (XL-MS). As the field surrounding XL-MS expands, there has been a rapid growth of analytical pipelines and crosslink chemistries. In light of this, the authors establish new false discovery rate (FDR) estimation software (IMP-X-FDR) based on a new synthetic peptide mix to compare and contrast the efficacies of various analytical software packages (MeroX, MS Annika, XlinkX, pLink, and MaxLynx) using a diverse set of crosslinking chemistries (DSSO, DSBU, CDI, ADH, DHSO, DSBSO). The current peptide mixture expands on previous work from these authors, building a library of a total of 141 crosslinkable peptides from the bacterial ribosome (~2.5 more than their previous effort).

I commend the authors for making their data and code available through PRIDE and GitHub. Additionally, the comprehensive application of crosslinker chemistries and analytical workflows is very interesting. Particularly, the finding that use of multiple crosslinking search engines to determine an intersection produces the most robust final dataset. These things may be conceptually straightforward but having evidence to this effect is important for the field. For example, the assertion that analyses should use a minimum peptide length of 6-7 and the benefits of enrichable crosslinks. There are however some issues with descriptions of the underlying code and methods that need to be addressed prior to publication.

Major Comments

1. Novelty of peptides. While the analytical comparisons are impressive and I believe important for the field, it is somewhat lost on me how relevant this relatively small number of peptides can recapitulate and draw strong optimization experiments for each of these pipelines. The core to this peptide mix seems to be analysis of a single protein complex. This is based on the small database size and number of peptides. Do the conclusions drawn from these peptides hold for different analyses, such as whole proteome crosslinking which XlinkX has been shown to be valuable for datasets with hundreds or thousands of crosslinks (Lui et al.

<https://doi.org/10.1038/ncomms15473>)?

2. The authors conclude that “The overlap of incorrect crosslinks is very low, showing that accumulating IDs from several replicate measurements to boost XL numbers is prone to also accumulate wrong hits, and should therefore be avoided.” Based on their findings though, the intersection of replicate analyses should provide high-confidence CSMs which would be of interest to many studies rather than concluding that replicates are wholly unhelpful. For Figure 1B, it would be helpful if the authors provided FDRs for the Venn diagram as in Figure 1C.

3. Though replicates seem to have been collected for many of these experiments, there is nearly no mention of any significance testing to back up claims of better/worse performance of these analyses. For example, is the difference in Figure 2A between DSSO and DSBU actually significant? For the DSSO versus DSBU analysis are the differences this claim is based on statistically significant when comparing replicate analyses? “Annika performs very well with DSSO and scores DSSO

crosslinks better than DSBU links (average score 279 for all DSSO links vs 269 for all DSBU links from our main library).”

4. IMP-X-FDR. The only explanation of the FDR recalculation seems to be this sentence: “It enables to calculate the real FDR based on crosslinks only allowed as correct in case they are formed within the same XL group...” Can the authors describe (through simplified formulae) what this program is doing compared to previous FDR estimation methods? For example, in Walztheoni et al. 2012:

<https://doi.org/10.1038/nmeth.2103>. Furthermore, there are very few comments within the source code and a very short README document. While the included manual is helpful it is more to help users runs experiments not understand what IMP-X-FDR is doing. Understanding this will be essential for assessing whether the claims such as “...IMP-X-FDR can be easily adopted for automated FDR calculation with any novel XL search engine...”. Additionally, in Figure 4B, it seems that peptide length plays a major role in FDR. Can IMP-X-FDR weight the recalculation and filtering by peptide length?

5. In Figure 3, many of the final FDR values are 0.0%. Thus, the FDR estimation seems as if it may be overly conservative for the ADH and DHSO analyses. There also seems to be a good deal of variance in the final number of unique crosslinked sites, which seems surprising owing to he controlled system and analytical pipelines. Can the authors comment on this?

6. Sites versus crosslinks. The figures in the manuscript switch between “# unique XLs” and “# unique XL sites” (Figure 4A vs 4B for example). Are these the same or are unique XLs considered as unique peptide linkages?

7. Database size. Figure 5 presents an interesting perspective of the effects of database size and complex backgrounds. It would however be interesting to see the effect of just one of these variables. Have the authors performed a similar analysis with just database size to determine if the effects observed in Figure 5 are more an affect of the database or the mixture? Similarly, with the search databases of only contained 171 proteins; how much does this really recapitulate a typical use case for the analytical pipelines at play? How much would this affect the assumptions or results of the FDR analyses performed?

8. In Figure 7B, the library peptides seem to all elute in the first half of the gradient. Do the uncrosslinked peptides elute in a similar pattern and show a similar retention effect as the crosslinked peptides? To the authors claim that they observed a retention time (RT) effect with more input mass, are these new CSM precursors present and not matched at higher RT (e.g., can they be seen in these analyses in an XIC)? It would also be helpful if the authors could show in the supporting information that “...the retention time of individual crosslink-sequence-matches (CSMs) did not change...”.

9. Do the authors have a sense of why: “MS Annika and MaxLynx seem to find this optimum best for cleavable crosslinkers, while pLink seems to perform very well for the tested non-cleavable linker.”?

10. FAIMS. The authors mention improved signal to noise ratios when using FAIMS, can they add these findings to Figure S2 as I believe it would be helpful data for the field?

Minor Comments

1. Shortening “crosslinked” to “XLed” is unnecessary in the first paragraph of the Results section.
2. Lysine is misspelled several times as “lysin”. There are a good number of minor grammatical errors throughout the text. Again, they are largely minor but can cause some confusion. E.g., p7;line 200 “we found less peptides” should be “we found fewer peptides”.
3. A callout in Figure 5 that the dataset is based on DSSO would be appreciated (in keeping with previous figures).
4. “Our results suggest that the choice of a properly sized database is of high importance...” The Bruce group previously established something similar which should be referenced:
<https://doi.org/10.1021/pr3011638>

REVIEWER

COMMENTS

Reviewer #1 (Remarks to the Author):

The manuscripts from the Mechtler group expands upon a previously published library of crosslinked peptides intended for benchmarking and optimization of CLMS analytic strategies. Namely it provides a means of estimating the true FDR of an analytic platform and enables the evaluation of the platforms ability to accurately assess FDR and a means to compare the performance of different strategies against the same benchmark. The main differences from the previous publication (Beveridge et al ref 22), is that this library is larger (1018 theoretical crosslinks vs 426 in previous version) and they have included a library suitable for benchmarking reagents that target acidic residues. An additional library (or really just a subset of the main one) is compatible with azide-containing crosslinker, commonly used in click based affinity enrichments. Finally, they also include a software tool that calculates FDR as well as physiochemical properties of crosslinked peptides to compare against theoretical distributions as a means to check the reliability of results.

I work in this field and have found the previous library to be an extremely valuable resource for developing and testing our FDR estimation software, and this newer library will no doubt continue to be a useful resource to people working in the field. I question the novelty of this particular paper, since it is just a minor expansion of the same principle as before. However, I will let the editor decide if the novelty is sufficiently to meet Nature Communications standards.

We thank the reviewer for carefully going through our manuscript and are happy that the (extended) peptide library will be a valuable resource for their future work as well.

The main downsides I see to this work are that the library only focuses on cleavable crosslinkers acquired using MS2 based strategies. There have been some recent publications (for instance Yugandhar, MCP 2019) that have tried to do similar work and found that MS2 only strategies were far less reliable than those incorporating MS3. Hence, it would be desirable to have the same library acquired with MS3 based acquisition instead of just using MS2 as done here. Even, if the FDR can be controlled by MS2 strategies, as shown here with the larger DB searches containing HEK spike-ins, it would be great to compare the performance and trade-offs of MS3 based approaches by acquiring the library using MS3, or at least, to enable others to do so. Additionally, it is a bit disappointing that they do not include results for BS3 acquired using a simple fixed HCD energy as this is still an incredibly useful and common approach. However, this is less of an issue since people developing these search engines can still use the previous library, and the MS2 results should still be applicable here (for instance pLink is tested).

We apologize for not including non-cleavable crosslinkers like BS3 in our current peptide library. We want to point in this regard to the previous paper from Beveridge et al., Nature Communications, 2020, that addressed already peptide library-based FDR estimations for non-cleavable crosslinkers. To avoid overlap between both papers we focused here on cleavable crosslinkers.

Of note, we included data of ADH as a non-cleavable crosslinker in our manuscript.

An additional omission is that there is no discussion of site-localization or benchmarking site-localization strategies. For Lys directed CLMS this is not typically an issue since essentially all peptides have a single modifiable Lysine residue, but it is a problem for the acidic-CLMS library. The strategy here seems to be to skirt the issue by creating peptides with only one possible site of modification, but this isn't a realistic scenario as most peptides will have multiple D/E residues.

Currently our peptide library is not designed to benchmark site localizations as each peptide contains just one linkable amino acid (K, D, E ect). Indeed, our peptide library does not represent natural sample since our peptides are synthesized but it does fulfil the purpose of checking the reported FDR within each software tool.

It is not possible to experimentally check for the right localization with our current library setup, but we reanalyzed the data allowing also STY for the lysin crosslinking and analyzed the data for DSSO. We could observe that searching with KSTY for lysine crosslinker does not necessarily increase or decrease the total number of crosslinks (except for plink2) but rather prioritise STY over K whenever it benefits the spectra description. The proportion of KSTY hits increases when convoluting CSMs to unique residue pairs since K-K links are represented through a higher number of K-K CSMs. Although the reaction specificity is low for STY compared to K, it is still present as possibility for a right linkage site which is why we now included it to an additional search. So far, we cannot check for the accuracy of the localisation data with the current peptide library. Of note, in the default settings of xiSearch a score penalty is given for non-K-K linkage sites to consider the higher nucleophilicity of primary amines making them to the preferred linkage site. To ensure a fair comparison to other tools we removed this score penalty.

This Figure is now part of the supplement.

Minor-

Figure 1: Clearly label or caption the figure to make clear that data were acquired by MS2 centric strategies.

They test pLink2, but the text refers to pLink. I would use pLink2 in the main text, at least at the first mention, for clarity. It is interesting that pLink2 performed best in the previous BS3 benchmarking study. This is worth discussing that pLink2 is using an open-modification strategy, while the other software are pre-determining the individual peptide masses based on the diagnostic ions, so quite a different algorithm. Also, statements in the text that pLink wasn't optimized for HCD don't make sense (pg 101)... pLink is primarily used with HCD data.

In the revised manuscript we added respective labels to clarify which acquisition strategy was used and changed pLink to pLink2. We agree that pLink2 is primarily used for HCD data in case of conventional crosslinking (non-cleavable), however this is not the case for MS-cleavable linkers. This we also confirmed with the authors of pLink2. In the selection menu of pLink version 2.3.9, that was the newest when preparing this manuscript, the available strategies showed only a CID-ETD option for MS-cleavable linkers (see figure below). In the current version 2.3.10 the cleavable linker option was entirely removed. We adopted the description in the paragraph "Benchmarking XL search engines with linkers targeting lysin." to avoid confusion.

We further agree that pLink 2 performed much better in the previous benchmark using BS3, which is a non-MS-cleavable linker. In the current study we benchmark using the MS-cleavable DSSO linker recorded using a stepped HCD acquisition strategy. This leads to a disadvantageous effect for our DSSO data using pLink 2 and the above reasons are included in our discussion. Of note, also in this study pLink 2 performs very well with the non-cleavable ADH linker (Figure 3 A) which strengthens our recommendation in the discussion part to use it for the analysis of non-cleavable but not for MS-cleavable linkers.

In panels B/C they show overlap in unique XL ids between different experimental replicates searched with Annika (B) and the same replicate searched with different search engines (C). They show in C) that FDR is low for the area of overlap compared to XLinks IDed by just one search engine. I would like to know if this is also true in panel B). how much better is FDR in XLs identified twice, and is it any better to use two separate searches of one replicate rather than one algorithm on two replicates?

We thank for pointing to this interesting question! The short answer is yes, it is true for panel B as well. The FDR of the 423 commonly (overlapping) found links in the left Venn of panel B is 0 %. As indicated in the original manuscript, this panel exclusively shows those crosslinks within the same group. Those 2 common crosslinks in the right Venn of panel B in the original manuscript have a FDR of 100 %. From this the calculated overall FDR in the commonly found links across all 3 replicates is 0.47 % (2 false vs 423 correct ones). To avoid confusion with the Venn's in panel B, we decided to include a single Venn Diagram showing total XL numbers and real FDRs instead and adopted the description in the revised manuscript.

Figure 2: I would find it helpful to have chemical structures in the figure so that I don't have to look them up. For instance, I didn't understand why the enrichable library was being tested until I realized that DSBOS is an azide containing reagent. The caption is mislabeled at one point B/C are incorrectly referenced.

We thank for this valuable suggestion, corrected the figure caption, and added the chemical structures of relevant linkers to Figure 2 in the revised manuscript.

Figure 3: Seems to be missing a caption for panel C. The results in the supplemental Figure 3 exploring the effect of neighboring amino acids on acidic crosslinking are interesting. I would consider using a standard Logo plot in Sup Figure 3A, as I found their data visualization a little hard to understand. Why does it seem like Leucine enhances crosslinking while Isoleucine inhibits?

We apologize for the missing caption which was added to the revised manuscript. We generated a sequence plot as suggested (see below) but we found the current design better readable compared to a logo plot for this specific graph containing the normalized amino acid frequency of crosslinked peptides.

We however adopted Suppl. Figure 3A that now exclusively includes those amino acids that seem to influence the formation of a crosslink strongest. This makes the graph more compact and better readable.

It is hard to comment on why Leucine seems to enhance while Isoleucine (on a different relative position) seems to hinder crosslink formation without having a significant amount of additional data and more replicates. Our supplemental figure therefore gives only a rough estimate on potential effects, which also strengthens us in our decision to show a simplified version of the graph now only.

In line 199 of the text, I don't understand what they are trying to say with respect to selection of 3+ and higher precursors.

We added some additional lines for clarification explaining why we used this MS method. In short: For crosslink data acquisition we use a DDA method that selects only +3 - +8 charged ions. With this we do not lose time for fragmentation of (predominantly +2 charged) linear peptides and instead focus on (predominantly +3/+4 charged) crosslinked peptides. To compare our non-crosslinked reference measurement with the amino acid distributions obtained in the crosslinked samples we used the same DDA method for crosslinked samples and non-crosslinked reference.

Figure 4: I'm surprised there is not much effect from setting separate intra-protein and inter-protein crosslink thresholds. I typically see many more intra-protein hits at the expense of fewer inter-protein hits. It would be useful if they reported stratified results like this. Also, the magnitude of this effect could depend a lot on the size of the data base being searched. I think it would be more informative to perform this analysis on the HEK spike in sample.

We agree with the reviewer and we usually also see many more intra protein hits in "real" proteome wide crosslinked samples. As suggested, we stratified our results and now report inter-/intra-protein crosslinks separately in our updated figure 4 and extended the related result-discussion in the manuscript. In contrast to our own and the reviewers experience from real samples, the relative number of inter-protein crosslinks is much higher compared to intra-protein crosslinks. This is also in line with the theoretical distribution expected based on the nature of the synthetic library originating for 38 different real protein sequences. As now discussed in the result part this serves as an additional hypothesis for the unexpected result of finding no strong effect in turning separate FDR calculation on or off in the respective search engines.

As suggested, we additionally investigated the effect in a larger database using our existing data (from figure 5) of the HEK spiked sample and analyzed it against the E. coli Ribosome + full human proteome (excluding human ribosomal proteins). The results are now shown in Supplemental Figure 1. Again, we found little to no effect on the resulting IDs and error rate in dependence of the chosen FDR calculation strategy. With XlinkX however, the number of identified (correct & incorrect) crosslinks

increased with separate FDR set to on leading to an overall experimentally validated FDR maintained at the same level of 18 – 20 %.

Figure 5/6 are nice.

Figure 7: I am surprised they mostly see 3+ precursors with DSSO -FAIMS. Is this reporting the charge of the CSMs (as indicated in the y-axis label) or all spectral matches (including linear peptides?). Figure S2 would be improved by addition of no-FAIMS data, but apparently they only collect this on the HFX and not the Exploris? It's not that hard to take off the FAIMS interface.

Yes, this is reporting the charge of CSM matches excluding linear peptides or peptides modified with the crosslinker (i.e. dead-end links). Within our peptide library system, we reproducibly find this charge distribution without FAIMS on the HFX. Charge distributions of CSMs were already investigated by Giese et al., *Mol Cell Proteomics*, 2016. In their figure 1 B they also report +3 charged (crosslink) precursors more abundant than +4 charged, while +2 charged species were most abundant for linear peptides, without FAIMS.

We performed an additional experiment on the Exploris to enable the comparison to data recorded without FAIMS on the same machine. We added this data to Supplemental Figure 8 (as also suggested by reviewer 3) in the updated manuscript and further updated figure 7 and its description accordingly. We still see a shift of relative charge distribution towards more +4 using FAIMS and more +5 with increased input. Indeed, we also do see a machine dependent effect as on the Exploris without FAIMS +3 and +4 charged precursors are roughly equally abundant, while on the HFX the +3 charged CSMs are more predominant (figure 7F).

Reviewer #2 (Remarks to the Author):

In this work, Matzinger et al presented an improved version of the synthetic peptide library for cross-linking mass spectrometry (XL-MS) studies (previously developed and published by their lab – Beveridge et al., Nature Communications, 2020). As with any high-throughput studies, quality-control is one of the most important aspects of XL-MS. Specifically over the past couple of years, some key papers have been published, demonstrating the existing quality-control issues in XL-MS along with potential solutions to address those (including the synthetic peptide library from the authors' own lab). Thus, establishing methods to carefully examine the quality of the results (including the development of the synthetic peptide library by the authors) is important for the rapidly evolving XL-MS field. However, the current manuscript is mainly an improvement on their previous synthetic peptide library (Beveridge et al. 2020) and might not be significant enough for a new publication especially in a journal of the high caliber as Nature Communications.

We thank the reviewer for carefully going through our manuscript and appreciating the importance of a solid quality control in the field of XL-MS. We agree that the idea of a peptide library is based on our previous publications by Beveridge et al. but we believe that the improvements made in size, complexity and useability are significant and therefore extremely valuable for the community.

Other specific comments:

1. As for benchmarking different acquisition strategies, I could not find any analysis for additional strategies such as MS2-MS2 and MS2-MS3 (with different types of energies; similar to the Figure 7c from their original benchmarking paper Beveridge et al., Nature Communications, 2020).

We recorded such data now and added it to figure 8 of the updated manuscript

2. Does their IMP-X-FDR tool allow 'FDR recalculation' at different levels (such as CSMs, residue pairs and protein pairs), similar to how the Xi FDR tool from Rappsilber lab allows for FDR estimation at different levels? If not, it would be a good functionality to add.

IMP-X-FDR allows checking of FDR calculations on residue pair level and as a new feature also CSM level but not on protein pair level.

3. The authors should compare the utility of their synthetic peptide library with that of existing quality assessment metrics for XL-MS, such as (i) the metric from Keller et al (Journal of Proteome Research, 2019) that accounts for the fraction of interprotein XLS and (ii) metrics from Yugandhar et al (Nature Methods, 2020) that accounts for corrected structure-based validation of XLS, along with metric utilizing additional false search spaces.

We added a respective comparison to the discussion part of our revised manuscript.

4. Adding a discussion on why different search tools perform better for specific linkers and specific acquisition strategies (software-specific results) would be informative. For example, their MS-Annika seems to be performing better on DSSO than DSBU (as described in the paper as well; Figure 2). Also, MeroX seems to perform drastically better on DHSO samples in comparison to ADH samples in terms of both sensitivity and specificity. Additionally, it is very interesting to note that in Figure 4A, XlinkX seems to perform drastically poorer when separate FDR calculation was turned on, which is not expected considering the advantage the separate FDR calculation has on the quality-control. I can understand that different search tools might be designed differently. However, drawing from the authors' group's own experience with development of cross-link search software, a thorough discussion relating the key technical aspects of cross-link identification with such soft-ware specific results would add value for this work.

We thank for raising the interesting and important questions. It is hard to determine the ultimate reasons for the performance differences but from our observations the following points might predominantly influence the linker specific performance:

- The level on which target-decoy based FDR estimation is performed (For MaxLynx, pLink and MeroX this is done on CSM level for all other tested tools the user can define on which level(s) FDR estimation is done)
- Some tools – on default- apply prescores to speed up data analysis. This however alters target-decoy based FDR estimation afterwards.
- Technical reasons due to altered spectra complexity for cleavable vs non cleavable linkers (see also reply to question 11 of reviewer 3)
- For the comparison of different X-Linker reagents the most important point seems the degree of compatibility of the search engine with the linker chemistry. Examples for this are already included to the discussion part of the manuscript: Proteome Discoverer does not fully support the correct definition of zero-length linkers as CDI, others lack an option to define more than two possible crosslinker fragments, which is relevant for sulfoxy based MS-cleavable linkers.
- In line with this observation, it seems that the developers of search engines focus on specific linker types for optimization of their algorithms and this yields in boosted results and better score-based separation of target vs decoy hits for linkers of the exact same chemistry.
- Regarding separate FDR calculation, the theoretical distribution of identified intra- vs inter-protein crosslinks in our library is similar to those within a real protein complex, however as all crosslinks are formed with equal probability the actual distribution of identified crosslinks is more in favor of interprotein crosslinks than in a real sample. This might induce unexpected results and in case of XlinkX the separate FDR calculation seems to fail -at least for our peptide library.

We extended our discussion part accordingly.

5. Minor: for efficient reproducibility of the results, please provide full protein sequences that were utilized as search databases for different analyses performed in this study in supplementary information.

Were added to PRIDE

Reviewer #3 (Remarks to the Author):

Nature Communications – 340243

The authors present a set of tools and software to benchmark analytical pipelines for cross-linking mass spectrometry (XL-MS). As the field surrounding XL-MS expands, there has been a rapid growth of analytical pipelines and crosslink chemistries. In light of this, the authors establish new false discovery rate (FDR) estimation software (IMP-X-FDR) based on a new synthetic peptide mix to compare and contrast the efficacies of various analytical software packages (MeroX, MS Annika, XlinkX, pLink, and MaxLynx) using a diverse set of crosslinking chemistries (DSSO, DSBU, CDI, ADH, DHSO, DSBSO). The current peptide mixture expands on previous work from these authors, building a library of a total of 141 crosslinkable peptides from the bacterial ribosome (~2.5 more than their previous effort).

I commend the authors for making their data and code available through PRIDE and GitHub. Additionally, the comprehensive application of crosslinker chemistries and analytical workflows is very interesting. Particularly, the finding that use of multiple crosslinking search engines to determine an intersection produces the most robust final dataset. These things may be conceptually straightforward but having evidence to this effect is important for the field. For example, the assertion that analyses should use a minimum peptide length of 6-7 and the benefits of enrichable crosslinks. There are however some issues with descriptions of the underlying code and methods that need to be addressed prior to publication.

We thank the reviewer for carefully going through our manuscript and appreciating comprehensiveness of our study and the importance of our findings! Please find below a point-to-point reply to your concerns.

Major Comments

1. Novelty of peptides. While the analytical comparisons are impressive and I believe important for the field, it is somewhat lost on me how relevant this relatively small number of peptides can recapitulate and draw strong optimization experiments for each of these pipelines. The core to this peptide mix seems to be analysis of a single protein complex. This is based on the small database size and number of peptides. Do the conclusions drawn from these peptides hold for different analyses, such as whole proteome crosslinking which XlinkX has been shown to be valuable for datasets with hundreds or thousands of crosslinks (Lui et al. <https://doi.org/10.1038/ncomms15473>)?

We agree that the number of peptides is relatively small compared to a real (proteome wide) sample. However, as already mentioned by the reviewer, our efforts to increase the size of the library are already significantly higher compared to any other published synthetic library. Furthermore, we added complexity not only by adding more peptides but also by allowing for more unique XL combinations (1018 possible unique links, which is on the same level as expectable results for complex protein/ in cell samples) and by allowing for inter- and intra-protein connections. The peptide sequences are designed from the *E.coli* ribosomal protein complex. We performed a shotgun analysis on this complex and found 171 proteins to be part of it. The thereby created database was also used for our searches within the manuscript.

In addition, our manuscript includes data of the peptide library spiked into a background of tryptic HEK peptides. While the purified synthetic peptides provide an ideal tool for method or software optimization, the spiked system mimics a real sample where the abundance of crosslinked peptides is low vs linear ones, and the matrix is complex. Like in a real proteome wide study we performed SEC

and affinity enrichment on those spiked systems. In conclusion, we believe that our overall dataset is comprehensive and large (by ID numbers) enough to take the drawn conclusions as basis for proteome wide studies. We believe that creating a synthetic system large enough to directly mimic a proteome wide system (without spiking) would be not feasible as the number of needed peptides would be tremendously higher.

2. The authors conclude that “The overlap of incorrect crosslinks is very low, showing that accumulating IDs from several replicate measurements to boost XL numbers is prone to also accumulate wrong hits, and should therefore be avoided.” Based on their findings though, the intersection of replicate analyses should provide high-confidence CSMs which would be of interest to many studies rather than concluding that replicates are wholly unhelpful. For Figure 1B, it would be helpful if the authors provided FDRs for the Venn diagram as in Figure 1C.

We thank for pointing to this interesting question! We would like to highlight that we fully agree with the reviewer’s opinion: Replicate measurements are helpful! It is however tricky to use (non-overlapping) XLs as they are prone to contain wrong IDs. Already in the original manuscript we state “A similar effect is also observed for replicate measurements (**Error! Reference source not found.**B). Of 425 unique crosslinks commonly found in three replicates only 2 (0.5 %) were incorrect.” within the section „Benchmarking XL search engines with linkers targeting lysin.“ Indeed, the FDR of the 423 commonly (overlapping) found links in the left Venn of panel B is 0 %. As indicated in the original manuscript, this panel exclusively shows those crosslinks within the same group. Those 2 common crosslinks in the right Venn of panel B in the original manuscript have a FDR of 100 %. From this the calculated overall FDR in the commonly found links across all 3 replicates is 0.47 % (2 false vs 423 correct ones). To avoid confusion with the Venn’s and in agreement with the suggestions of reviewer 1 and 3, we decided to include a single Venn Diagram showing total XL numbers and real FDRs instead in Figure 1B and adopted the description in the revised manuscript.

3. Though replicates seem to have been collected for many of these experiments, there is nearly no mention of any significance testing to back up claims of better/worse performance of these analyses. For example, is the difference in Figure 2A between DSSO and DSBU actually significant? For the DSSO versus DSBU analysis are the differences this claim is based on statistically significant when comparing replicate analyses? “Annika performs very well with DSSO and scores DSSO crosslinks better than DSBU links (average score 279 for all DSSO links vs 269 for all DSBU links from our main library).”

We now included statistics for Figure 2A and B and changed the text accordingly.

4. IMP-X-FDR. The only explanation of the FDR recalculation seems to be this sentence: “It enables to calculate the real FDR based on crosslinks only allowed as correct in case they are formed within the same XL group...” Can the authors describe (through simplified formulae) what this program is doing compared to previous FDR estimation methods? For example, in Walztheoni et al. 2012: <https://doi.org/10.1038/nmeth.2103>. Furthermore, there are very few comments within the source code and a very short ReadMe document. While the included manual is helpful it is more to help users runs experiments not understand what IMP-X-FDR is doing. Understanding this will be essential for assessing whether the claims such as “...IMP-X-FDR can be easily adopted for automated FDR calculation with any novel XL search engine...”.

IMP-X-FDR is not per se calculating a new FDR but rather checking whether the FDR estimated by the crosslink search engines specific FDR tools are correct or not. The user must provide a table containing only target-target crosslink hits as input file. Based on this file the IMP-X-FDR tool compares crosslink sequences to a peptide library template file containing all possible peptide combinations within each

group of peptides. The template file of the main peptide library contains 20 groups of peptides with known linkage sites. If a crosslinked peptide of the result file (file after tool specific search and FDR filtering) matches a crosslinked peptide within one group of possible crosslinks, its counted as a true hit. If it matches one peptide of group A and one of group B its counted as false positive. After this checking step the tool counts all true and false hits and calculates an “experimentally validated FDR”.

We included a more detailed description of the process as well as a simplified formula in the revised manuscript and extended our readme file on github.

5. Additionally, in Figure 4B, it seems that peptide length plays a major role in FDR. Can IMP-X-FDR weight the recalculation and filtering by peptide length?

No, IMP-X-FDR does not weight or filter FDR dependent on peptide length. To create the results shown in Figure 4, we performed separate data-analyses with different minimal peptide lengths allowed in the search algorithm. After that we checked the resulting experimentally validated FDR on each result file using IMP-X-FDR. However, a manual filtering for peptide lengths of interest is possible in the output file of IMP-X-FDR. By then counting correct and incorrect IDs (as marked in the output file) the FDR of any subpopulation within existing resultfiles can be calculated ($\frac{\text{\#incorrect IDs}}{\text{\#total IDs}}$).

6. In Figure 3, many of the final FDR values are 0.0%. Thus, the FDR estimation seems as if it may be overly conservative for the ADH and DHSO analyses. There also seems to be a good deal of variance in the final number of unique crosslinked sites, which seems surprising owing to he controlled system and analytical pipelines. Can the authors comment on this?

Figure 3 shows the number of crosslink IDs from ADH and DHSO after analysis with different search algorithms. All of them estimate their FDR using a target-decoy approach. Using IMP-X-FDR we checked based on the experimental design of our library, how many of the reported crosslinks on 1% FDR level are correct (within the same peptidegroup) or incorrect (crosslink across groups). Based on this we calculated the experimentally validated FDR (reported incorrect IDs/total number of IDs). This result is shown in the bars with a solid line, and all bars show a “real” FDR level >0% and also > the expected 1%. Therefore, we applied a post-score cutoff until we reached a real FDR smaller or equal to 1% (bars with dashed lines). Since the final number of crosslink IDs is <100 in all cases of Figure 3 A and B we did not allow a single incorrect crosslink to be contained in the data. This results in the shown 0% FDR for all bars with dashed lines.

The small ID numbers are most likely also responsible for the high variance across replicates in this case. Variance is clearly lower in graphs with higher ID numbers like panel C of Figure 3 or also the results of Figure 1 and 2. This shows the peptide library does indeed deliver quite reproducible results compared to real samples. Of note, replicates were measured on different days which is why some variance occurs also from altered day-to-day performance on our masspecs.

7. Sites versus crosslinks. The figures in the manuscript switch between “# unique XLs” and “# unique XL sites” (Figure 4A vs 4B for example). Are these the same or are unique XLs considered as unique peptide linkages?

Yes, these are the same and they correspond to unique residue pairs (based on the proteins the synthetic peptide sequences are originating from). We apologize for the confusion and use a uniform nomenclature in the revised manuscript.

8. Database size. Figure 5 presents an interesting perspective of the effects of database size and complex backgrounds. It would however be interesting to see the effect of just one of these variables. Have the authors performed a similar analysis with just database size to determine

if the effects observed in Figure 5 are more an affect of the database or the mixture? Similarly, with the search databases of only contained 171 proteins; how much does this really recapitulate a typical use case for the analytical pipelines at play? How much would this affect the assumptions or results of the FDR analyses performed?

We agree that this is an effect might not only come from database size but also from the sample complexity.

Effect of sample complexity: We analyzed the spiked, the SEC enriched (both figure 5) as well as the clean crosslinked peptidelibrary (figure 1) using our 171 protein database. When comparing these results, we see an increase in FDR for some tools (i.e. MeroX 5.7% to 9.2% and 6.2 %; XlinkX 4.4% to 6.7% and 12.1% for clean library vs spiked vs enriched respectively) and a constant or even lowered FDR other tools with increased sample complexity. (i.e. Annika 2.7% to 1.7% and 2.5%, pLink 4% to 4.1% and 4.8%, MaxLynx 2.2% to 0.5% and 3.3%, Xisearch 3.2% to 2% and 3.2% for clean library vs spiked vs enriched respectively). Based on these observations no clear conclusion can be drawn and this (small) differences in FDR might rather be random based than on sample complexity.

Effect of database size: We added an additional analysis of our non-spiked DSSO linked main library with Annika where we increased the database size up to proteome wide. The number of crosslink IDs drops while FDR slightly increases (see below and supplemental figure 2C in the updated manuscript), which is the same behavior as seen after spiking (figure 5).

We therefore conclude that our assumptions are more relevant in dependence of the database size used. This also affects real (proteomewide) studies as they involve large databases. Keeping the database size smaller sized might however be advantageous wherever possible.

9. In Figure 7B, the library peptides seem to all elute in the first half of the gradient. Do the uncrosslinked peptides elute in a similar pattern and show a similar retention effect as the crosslinked peptides?

The crosslink CSM distribution centers towards the middle of the gradient after MSON time correction. We apologize for the confusion and changed this accordingly in figure 7.

10. To the authors claim that they observed a retention time (RT) effect with more input mass, are these new CSM precursors present and not matched at higher RT (e.g., can they be seen in these analyses in an XIC)? It would also be helpful if the authors could show in the supporting information that "...the retention time of individual crosslink-sequence-matches (CSMs) did not change...".

The referred effect was observed based on identified CSMs only as only those that are not hidden in the noise and therefore fragmented and identified are relevant for the final result.

As suggested, we representatively checked this for 8 crosslinked peptides that were identified across several injection amounts. (No CSM was found across all files). We included this result as an additional supplemental figure 8B and below show that indeed no shift in RT of individual CSMs is observable.

11. Do the authors have a sense of why: “MS Annika and MaxLynx seem to find this optimum best for cleavable crosslinkers, while pLink seems to perform very well for the tested non-cleavable linker.”?

We think the reason for this observation is rather coming from the MS spectra than from the crosslinker itself. Cleavable crosslinker result in more complex spectra than non-cleavable ones in terms of more fragment-ions. If a programmer optimizes their software tool than based on these spectra and on the chosen scoring function of the fragments or quality of the spectra. Since each crosslinking group specialized or started with either non or cleavable crosslinkers the software is heavily influenced by these starting conditions. So, we think the differences come from a technical reason rather than a chemical. Please also refer to our reply to question 4 of reviewer 2 on that topic.

12. FAIMS. The authors mention improved signal to noise ratios when using FAIMS, can they add these findings to Figure S2 as I believe it would be helpful data for the field?

We directly compared our crosslinked library with vs without FAIMS on the same machine and added the result to the respective figure (Supplemental Figure 3/panel C and D in the updated manuscript).

Below we also add a direct comparison of the signal to noise when integrating all peaks using Thermo Free Style. In line with other literature (e.g. Swearingen K. E. et al. Expert Rev. Proteomics, 505–517

(2012); Gerbasi, R. et al. (2021) doi:10.26434/chemrxiv.13653578.v1.) the S/N ratio is improved by factor 1.5 – 4 by FAIMS (see figure below).

Please also refer to our answer to Reviewer 1/Figure 7 on a similar topic.

Minor Comments

1. Shortening “crosslinked” to “XLed” is unnecessary in the first paragraph of the Results section.

We changed this in the revised manuscript.

2. Lysine is misspelled several times as “lysin”. There are a good number of minor grammatical errors throughout the text. Again, they are largely minor but can cause some confusion. E.g., p7;line 200 “we found less peptides” should be “we found fewer peptides”.

Thank you for pointing to these typos. We corrected them and read through the entire manuscript for correction of other typos.

3. A callout in Figure 5 that the dataset is based on DSSO would be appreciated (in keeping with previous figures).

Changed

4. “Our results suggest that the choice of a properly sized database is of high importance...”
The Bruce group previously established something similar which should be referenced:
<https://doi.org/10.1021/pr3011638>

We added the work of the Bruce group to our results section dealing with the variation of database size.

REVIEWERS' COMMENTS

Reviewer #3 (Remarks to the Author):

The authors have addressed all of my main concerns. I believe the software, peptide library, and comprehensive analyses will be very valuable for the XL-MS field.

Minor:

- Figure 1B. The overlap between Rep 1 and Rep 2 is missing an FDR estimate. This may be minimally informative but should be included for completeness.
- P15, line 422. "Of note, not all tested tools allow to perform their target-decoy..."
 - o This should be reworded or edited for clarity.
- P15, line 432. "...and is therefore an allrounder within the crosslink search engines."
 - o This is a confusing sentence and should be reworded.

REVIEWERS' COMMENTS

Reviewer #3 (Remarks to the Author):

The authors have addressed all of my main concerns. I believe the software, peptide library, and comprehensive analyses will be very valuable for the XL-MS field.

We would like to thank the reviewer for again going carefully through our manuscript and the positive feedback!

Minor:

- Figure 1B. The overlap between Rep 1 and Rep 2 is missing an FDR estimate. This may be minimally informative but should be included for completeness.

In that overlap no false crosslinks were found, and we annotated this now as 0% FDR in the revised manuscript.

- P15, line 422. "Of note, not all tested tools allow to perform their target-decoy..."
 - o This should be reworded or edited for clarity.

We edited for more clarity.

- P15, line 432. "...and is therefore an allrounder within the crosslink search engines."
 - o This is a confusing sentence and should be reworded.

We reworded for more clarity.